# Cortical and subcortical signatures of conscious object recognition

Max Levinson [1,5], Ella Podvalny[1,5], Steven H. Baete [2] & Biyu J. He [1,2,3,4 ✉]

The neural mechanisms underlying conscious recognition remain unclear, particularly the roles played by the prefrontal cortex, deactivated brain areas and subcortical regions. We investigated neural activity during conscious object recognition using 7 Tesla fMRI while human participants viewed object images presented at liminal contrasts. Here, we show both recognized and unrecognized images recruit widely distributed cortical and subcortical regions; however, recognized images elicit enhanced activation of visual, frontoparietal, and subcortical networks and stronger deactivation of the default-mode network. For recognized images, object category information can be decoded from all of the involved cortical networks but not from subcortical regions. Phase-scrambled images trigger strong involvement of inferior frontal junction, anterior cingulate cortex and default-mode network, implicating these regions in inferential processing under increased uncertainty. Our results indicate that content-specific activity in both activated and deactivated cortical networks and non-content-specific subcortical activity support conscious recognition.

[1] Neuroscience Institute, New York University School of Medicine, New York, NY, USA. [2] Department of Radiology, New York University School of Medicine, New York, NY, USA. [3] Department of Neurology, New York University School of Medicine, New York, NY, USA. [4] Department of Neuroscience & Physiology, New York University School of Medicine, New York, NY, USA. [5] These authors contributed equally: Max Levinson, Ella Podvalny. ✉email: biyu.he@nyulangone.org

The neural mechanisms of conscious perception remain a lasting mystery in neuroscience[1,2]. Despite intense research, there is currently no general consensus on the brain regions involved in supporting the content of conscious experience. Several leading theories make different, experimentally testable predictions about neural activity underlying conscious perception as compared to unconscious processing. Recurrent Processing Theory (RPT)[3] suggests that conscious perception results from recurrent neural activity that both encodes and integrates features of the stimulus perceived. According to RPT, such recurrent processes can be implemented by sensory pathways alone, such as the connections within and between early and high-order visual cortices. Information Integration Theory (IIT)[4] argues for differentiated, yet integrated, information as a hallmark of conscious experience and proposes a 'posterior hot zone' including occipitotemporal sensory areas and parietal cortex as the content-specific neural correlate of conscious perception. By contrast, Global Neuronal Workspace (GNW)[5] theory predicts that encoding of stimulus content in posterior sensory pathways alone cannot generate conscious experience: additional propagation of stimulus content information across a highly interconnected large-scale brain network composed of frontal and parietal areas is necessary.

All theories agree that neural activity encoding stimulus content in sensory areas is a prerequisite for conscious experience in humans—a claim that is supported by abundant experimental evidence[6–8]. The divergence between theories centers around whether large-scale broadcasting of stimulus content beyond sensory networks is involved[5]. The experimental evidence supporting this key prediction of the GNW theory, however, is scarce. Most evidence supporting GNW was obtained by contrasting neural responses between 'seen' and 'unseen' trials[9–11] or between hit/false-alarm and miss/correct-rejection trials[12], but see[13]. While such evidence is valuable, it is insufficient to support the hypothesized content-specific broadcast. Recent studies investigated the neural correlates of conscious content using multivariate pattern decoding applied to magnetoencephalography (MEG) data[14,15], but this technique cannot definitively discern the neuroanatomical origins of decoded information[16].

Surprisingly, none of these leading theories of consciousness make specific predictions concerning several brain structures key to information flow. First, the default-mode network (DMN) is a hub of corticocortical communication[17,18] and anchors one end of the large-scale cortical gradient, situated opposite to primary sensory areas[19]. The DMN has so far been overlooked in studies of perception, primarily because it typically deactivates during externally driven sensory tasks[20]. However, recent findings show that its activity patterns during deactivation in fact contain information about the content of visual perception[21,22]. Studies of perceptual processing in the DMN remain sparse, and it is currently unknown whether the DMN is involved in conscious recognition. Second, subcortical structures including basal ganglia[23,24], thalamus[25] and brainstem[26,27] contribute to large-scale cortical communication and influence sensory processing and perceptual decision-making. While it is well established that thalamus and brainstem regulate arousal and enable conscious wakefulness[28–31], the precise role that these regions play in conscious perception, within the wakeful state, remains poorly understood[5,32].

The choice of stimuli and task paradigm may introduce another source of variability when interpreting studies of conscious perception. Everyday perceptual tasks involve recognizing and interacting with objects and scenes. By contrast, visual consciousness research often resorts to low-level visual stimuli, such as Gabor patches, for enhanced experimental control. Awareness of low-level features (e.g., luminance changes) can contribute to unconscious processing of high-level visual information (e.g., a face)[33], but conscious perception of a high-level object is not reducible to conscious perception of low-level features and may involve additional/different mechanisms. Furthermore, the choice of technique to render the stimulus invisible can influence observed neural activity. For example, the majority of previous studies on liminal object recognition have used visual masking to reduce stimulus visibility[9–11,34–36]. This approach may result in nonlinear interactions between object processing and mask processing, complicating interpretation of results. In addition, the mask may render the stimulus invisible by disrupting its processing at an early stage[37], which may prevent the propagation of activity across large-scale brain networks that occurs in more natural scenarios.

Here we shed light on the neural mechanisms underlying conscious object recognition across large-scale cortical and subcortical networks. We recorded whole-brain blood-oxygen-level-dependent (BOLD) activity using high-field (7 Tesla) functional magnetic resonance imaging (fMRI) while subjects performed a liminal object recognition task. Our findings reveal content-specific neural activity underlying conscious object recognition across both activated and deactivated cortical networks, together with non-content-specific activity in subcortical regions that facilitates conscious recognition. They point to a broader view of the neural correlates of conscious perception than suggested by existing theories, and underline the need to better understand the roles of default-mode network and subcortical regions.

## Results

**Paradigm and behavior.** To identify neural mechanisms underlying conscious object recognition, we designed an experimental paradigm wherein object stimuli are presented at a liminal contrast[38]. We operationalized "recognition" by instructing subjects to report whether they saw an object. Subjects were instructed to respond "yes" whenever they saw an object, even if the visibility was unclear, and respond "no" when they saw nothing or low-level features only, such as lines or cloud-like abstract patterns. This definition of subjective recognition, allowing unrecognized trials to include conscious perception of low-level features, is consistent with prior studies of conscious object recognition[34–36]. To identify the liminal image contrast, we conducted an adaptive staircase procedure whereby the contrast of each image was titrated (see Fig. 1a, b and Methods) to reach a ~50% subjective recognition rate across identical trials for each participant. The present paradigm is analogous to threshold-level visual detection tasks using low-level stimuli[39–41], but with important differences in stimulus type (e.g., Gabor patches vs. objects) and the definition of threshold (visibility of any stimulus feature vs. object recognition).

Experimental stimuli included four common visual object categories: faces, animals, houses and manmade objects (Fig. 1c). In each trial, participants reported the category (four-alternative choice discrimination) of the presented image and their recognition experience (Fig. 1d). If the image was not recognized, they were instructed to make a genuine guess about its category. The stimuli set included real and scrambled images. Scrambled images were created by phase-shuffling a randomly chosen real image from each category to preserve category-specific low-level image features but destroy any meaningful content (see Methods), and were presented at the same contrast as their real object counterparts. The stimuli were presented in a randomized order to prevent category predictability.

Participants reported 48.0 ± 2.6% (mean ± s.e.m., $N = 25$) of real images as recognized (i.e., % of "yes" reports), which did not differ from the intended recognition rate of 50% (Fig. 1f, Wilcoxon signed-rank test, $W = 126.5$, $p = 0.33$). We computed

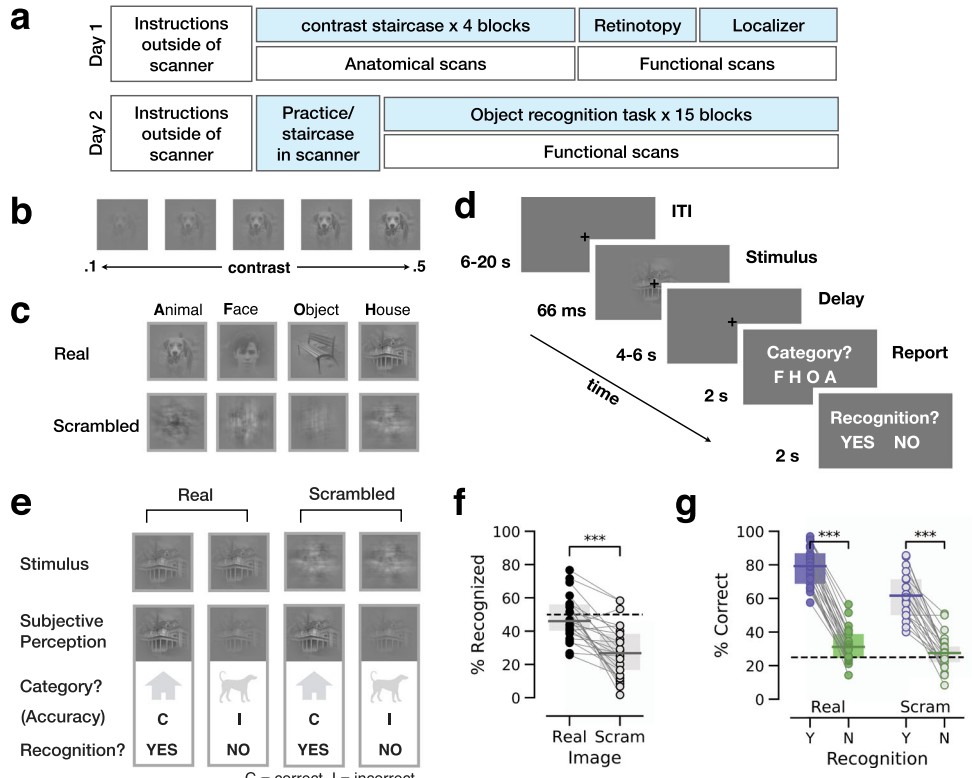

**Fig. 1 Experimental paradigm and behavior. a** A schematic timeline of experimental sessions conducted across two days. **b** Example of an object image presented at various levels of contrast. **c** Examples of real and scrambled object images from four categories. **d** Trial structure of the main object recognition task. Following a central fixation period, participants viewed a brief stimulus presented at an individually-calibrated recognition threshold contrast. Next, participants reported the stimulus category and their recognition experience. **e** Illustration of trial types. Each participant may report the same stimulus (top) as recognized in one trial and as not recognized in another. **f** Percentage of trials reported as recognized for real and scrambled images (N = 25). Dashed line indicates intended threshold-level recognition rate for real images (two-sided Wilcoxon sign-rank test). ***$p = 4.1 \times 10^{-5}$.
**g** Accuracy of category report in recognized and unrecognized image trials for real (N = 25, ***$p = 1.2 \times 10^{-5}$) and scrambled (N = 22, only subjects reporting more than 5 scrambled-image trials as recognized are included, ***$p = 5.96 \times 10^{-5}$) images (two-sided Wilcoxon sign-rank test). The accuracy of reported category in scrambled-image trials is determined based on the category of the original stimulus used to generate the scrambled image. In all figures, boxplot center line depicts the median and bounds indicate upper and lower quartiles.

every subject's recognition rates for each individual object image (Supplementary Fig. 1a). Subjects recognized the majority of object images at a rate of 46.8 ± 4.7% (mean ± s.e.m. of the mode, where each subject's mode is the center of their tallest histogram bin shown in Supplementary Fig. 1a), confirming success of our staircase procedure on the single-subject level. The recognition rate of scrambled images was 28.0 ± 3.1%, significantly below the recognition rate of real images (Fig. 1f, Wilcoxon signed-rank test, $W = 315.0$, $p = 4.1 \times 10^{-5}$) and significantly above zero (Wilcoxon signed-rank test, one-sided, $W = 325.0$, $p = 6.12 \times 10^{-6}$). Because participants were instructed to report recognition only if they perceived an object, "yes" reports to scrambled (noise) images constitute false alarms in a signal detection framework[38].

We next characterized categorization behavior in recognized and unrecognized trials. Categorization accuracy of real object images was 78.8 ± 2.2% for recognized images and 32.0 ± 1.9% for unrecognized images. Recognized images produced a significantly higher categorization accuracy than unrecognized images ($W = 325.0$, $p = 1.2 \times 10^{-5}$, $N = 25$, Wilcoxon signed-rank test, Fig. 1g). Categorization accuracy for unrecognized images was still significantly above the chance level of 25% ($W = 251.0$, $p = 3.9 \times 10^{-3}$, $N = 25$, Wilcoxon signed-rank test, Fig. 1g), suggesting that stimuli that failed to be consciously recognized were processed unconsciously to the point of influencing categorization behavior, consistent with earlier studies[42]. Category-specific categorization

behavior is shown in Supplementary Fig. 1b. This pattern of behavioral results is fully consistent with a previous study using the same paradigm carried out at a faster pace[38].

Interestingly, subjects' categorization behavior in scrambled image trials followed a similar pattern (Fig. 1g): categorization accuracy (assessed by whether the reported category corresponds to the category of the original image from which the scrambled image was generated) was 61.7 ± 2.8% for recognized images and 27.7 ± 2.2% for unrecognized images, which was significantly above chance only for recognized images (Wilcoxon signed-rank test, $W = 253.0$, $p = 3.95 \times 10^{-5}$ and $W = 159.0$, $p = 0.29$, respectively). Categorization accuracy was also significantly higher for recognized than unrecognized scrambled images (only participants with more than 5 scrambled images in each group were included, $N = 22$, Wilcoxon signed-rank test, $W = 231.0$, $p = 5.96 \times 10^{-5}$). These results are consistent with evidence suggesting that category-specific low-level image features, which are preserved in phase-scrambled images, contribute to categorization behavior[43]. Because "Yes" reports to scrambled images were accompanied by higher categorization accuracy than "No" reports, these false-alarm reports likely reflect genuine false perception as opposed to button press mistakes.

Lastly, we tested for any change in perceptual behavior across task blocks. Recognition rate and categorization accuracy did not change significantly over the course of the experiment (Supplementary Fig. 1b, c, one-way repeated-measures ANOVA,

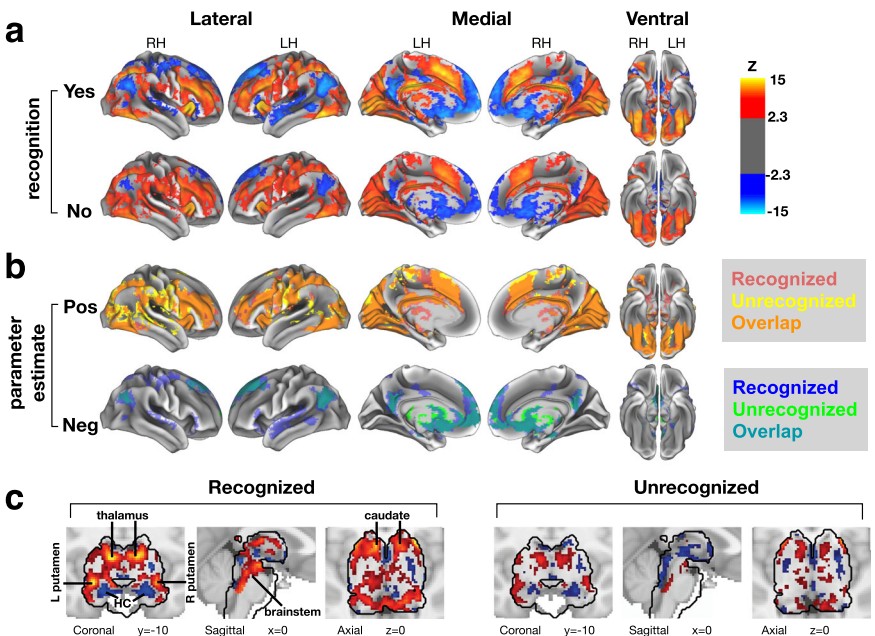

**Fig. 2 Widespread brain response to liminal object stimuli. a** Statistical parametric maps of BOLD response to liminal object stimuli that were subsequently recognized (Yes, top row) and unrecognized (No, bottom row), plotted on an inflated cortical surface template. **b** Same data as in **a**, showcasing the overlap in brain areas exhibiting positive ("pos") and negative ("neg") responses from baseline according to GLM parameter estimates. The positive and negative responses indicate activation and deactivation, respectively. **c** Same analysis as **a** presenting activity in subcortical regions only (within black contour), whereas the cortical activity shown in **a** is excluded for clarity (plotted on MNI152 template brain volume). The results are shown for voxels exhibiting significant *p*-values only, which were determined using cluster-based FWE-correction in FSL, with a voxel-defining Z statistic threshold of *p* < 0.01 and a cluster size threshold of *p* < 0.05.

$F_{14, 294} = 0.98$, $p > 0.4$ and $F_{14, 294} = 1.63$, $p > 0.07$, respectively), suggesting that there was no substantial learning or fatigue effect in our task. This was partly because subjects were already familiarized with the stimuli set from the staircasing and localizer tasks prior to the main task.

**Widespread brain responses to recognized and unrecognized object images.** Subjects' brain activity was recorded by whole-brain 7 T fMRI while they performed the aforementioned task. Using a general linear model (GLM), we extracted BOLD responses triggered by stimuli that were reported as recognized or unrecognized ("Yes"/"No"). The fMRI BOLD signal reflects local metabolism associated with neuronal activity and, accordingly, activation implies greater population neuronal activity and deactivation implies reduced neuronal activity[44]. We found widespread cortical responses (*p* < 0.05, FWE-corrected across the whole brain) to both recognized and unrecognized (real) object images (Fig. 2a). Stimulus-induced activation was found in areas previously implicated in object recognition, including occipito-temporal visual cortex[45] and orbitofrontal cortex[36]. In addition, we observed activation in frontoparietal 'task-positive' areas that were previously shown to correlate with conscious perception[9], as well as stimulus-induced deactivation in DMN regions and bilateral hippocampi, which are coupled with the DMN[46]. However, strikingly, the activation and deactivation responses span overlapping spatial extents in recognized and unrecognized trials (Fig. 2b).

Several subcortical regions also responded to object images (*p* < 0.05, FWE-corrected across the whole brain, Fig. 2c). In both recognized and unrecognized conditions we observed stimulus-induced activation in the brainstem and bilateral thalami. Sub-regions of the basal ganglia, including both ventral (nucleus accumbens) and dorsal (body of caudate, putamen) striatum, showed significant activation in recognized trials; among these,

only the body of caudate also exhibited significant activation in unrecognized trials.

The overlapping spatial extent of brain response between recognized and unrecognized trials is surprising and contrary to the GNW theory, which predicts that conscious perception involves recruitment of additional large-scale cortical networks that are largely silent during unconscious perception[5,47]. Our data suggest instead that threshold-level object stimuli elicit spatially similar widespread responses—across activated and deactivated large-scale brain networks—regardless of subjective recognition outcome. Subjective recognition may therefore reflect response magnitude and/or pattern differences within these distributed brain networks. We directly test this possibility in the following sections.

**Subjective object recognition correlates with amplified activation and deactivation.** While responses to recognized and unrecognized object images show similar spatial extent, the response magnitude varied with recognition status as revealed by a GLM contrast of recognized > unrecognized trials (Fig. 3a; *p* < 0.05, FWE-corrected; MNI coordinates of significant clusters listed in Supplementary Table 1). Almost all cortical and sub-cortical regions that responded to liminal object stimuli (Fig. 2a) also showed a difference in their response magnitudes to recognized vs. unrecognized images (Fig. 3a), including both activated and deactivated regions.

Does the recognition effect (Fig. 3a) stem from a change in response magnitude or a change in response direction (activation vs. deactivation)? To answer this question, we extracted regions-of-interest (ROIs) from the recognized > unrecognized contrast (see Methods, *ROI definition from recognition contrast*) and, for each ROI, calculated percent signal change of the BOLD response (Fig. 3b). This analysis shows that positive contrast values (yellow-red in Fig. 3a) stem from stronger activation in

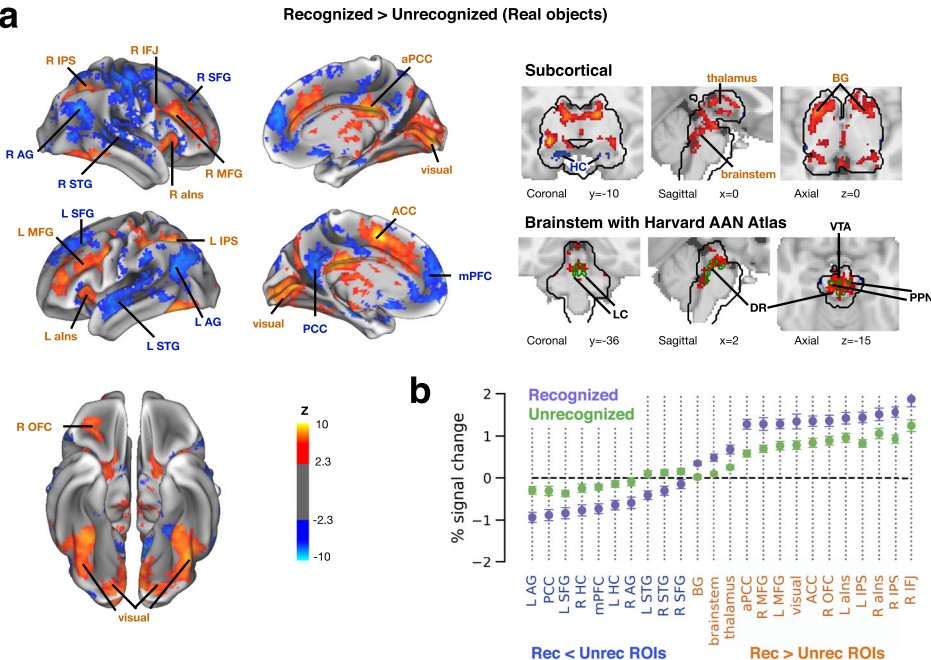

**Fig. 3 Effects of subjective object recognition on BOLD response magnitudes across the brain. a** Group-level GLM contrast of recognized > unrecognized objects. Left: data overlaid on a template cortical surface. Right, top: significant effects in subcortical regions (black contour). Right, bottom: significant effects in brainstem (black contour), overlaid with ROIs from the Harvard Ascending Arousal Network Atlas (green contours). Plotted on MNI152 template brain volume. Data were thresholded using cluster-based FWE-correction in FSL, with a voxel-defining Z statistic threshold of $p < 0.01$ and a cluster size threshold of $p < 0.05$. **b** GLM-estimated percent signal change of the stimulus-triggered response, averaged across voxels within significant clusters shown in A. Blue and orange labels denote clusters with negative and positive contrast values, respectively; ROIs are sorted according to response magnitudes in recognized trials. Data are sorted by subjective recognition report, averaged across stimulus categories. Statistical testing was not performed to avoid circularity. Error bars indicate s.e.m. across subjects ($N = 25$). BG: basal ganglia, IPS: intraparietal sulcus, aPCC: anterior posterior cingulate cortex, ACC: anterior cingulate cortex, aIns: anterior insula, IFJ: inferior frontal junction, MFG: middle frontal gyrus, OFC: orbitofrontal cortex, HC: hippocampus, AG: angular gyrus, PCC: posterior cingulate cortex, STG: superior temporal gyrus, SFG: superior frontal gyrus, mPFC: medial prefrontal cortex, LC: locus coeruleus, DR: dorsal raphe, VTA: ventral tegmental area, PPN: pedunculopontine nucleus, L: left, R: right.

recognized trials (Fig. 3b, right) in subcortical (brainstem, thalamus, basal ganglia), visual and frontoparietal 'task-positive' networks; negative contrast values (blue in Fig. 3a) stem from stronger deactivation in recognized trials (Fig. 3b, left) in the DMN and bilateral hippocampi.

These results indicate that conscious object recognition is associated with stronger positive and negative responses across widely distributed brain networks. They are consistent with previous visual masking studies showing recognition-related effects in ventral temporal cortex[45], orbitofrontal cortex[36], and lateral frontoparietal regions[9] but also reveal additional brain regions whose activity correlates with subjective recognition, including anterior cingulate cortex, bilateral insulae, DMN, and subcortical structures.

**Recognition-related effects in subcortical brain regions**. We performed several analyses to better understand the recognition-related effects in subcortical regions revealed by the above analysis. First, to pinpoint the source of thalamic activation related to recognition, we entered the MNI coordinates of each local maximum from both left and right thalamus (Supplementary Table 1) into the Thalamic Connectivity Atlas[48] which outputs the probability of anatomical connectivity between a given thalamus voxel and several broad cortical areas. Both voxels had the highest probability of connectivity with prefrontal cortex compared to other cortical areas (left: 43%, right: 73%). The primary projection from the thalamus to prefrontal cortex is from the mediodorsal nucleus (MD)[49], suggesting that the differential thalamic

activation during recognized vs. unrecognized trials likely originated from the MD nucleus.

We next investigated whether the lateral geniculate nucleus (LGN), the earliest hub of visual information processing in the brain, is also involved in this task. At previously published MNI coordinates for the LGN[50] (Supplementary Table 2) we observed a cluster of voxels showing a significant difference in response to recognized vs. unrecognized object images in each hemisphere ($p < 0.05$, FWE-corrected across the whole brain), supporting a correlation between LGN activity (which could reflect feedback from V1) and subjective recognition[51].

Lastly, we overlaid the Harvard Ascending Arousal Network Atlas[31] on the recognized > unrecognized activity contrast (Fig. 3a, right) to better identify brainstem structures. We found an effect in pedunculopontine nucleus (PPN) and locus coeruleus (LC), two nuclei situated in pons, and in ventral tegmental area (VTA) and dorsal raphe nucleus situated in the midbrain. MNI coordinates for identified brainstem structures are presented in Supplementary Table 2. As components of the arousal network, activity in these areas may reflect changes in arousal that correlate with or predict[38] subjective recognition of liminal stimuli.

**The widespread cortical responses to recognized objects carry category information**. To test whether the amplified activation and deactivation during subjective recognition contained information about perceived stimulus content, we examined whether the object category could be decoded from activity patterns in

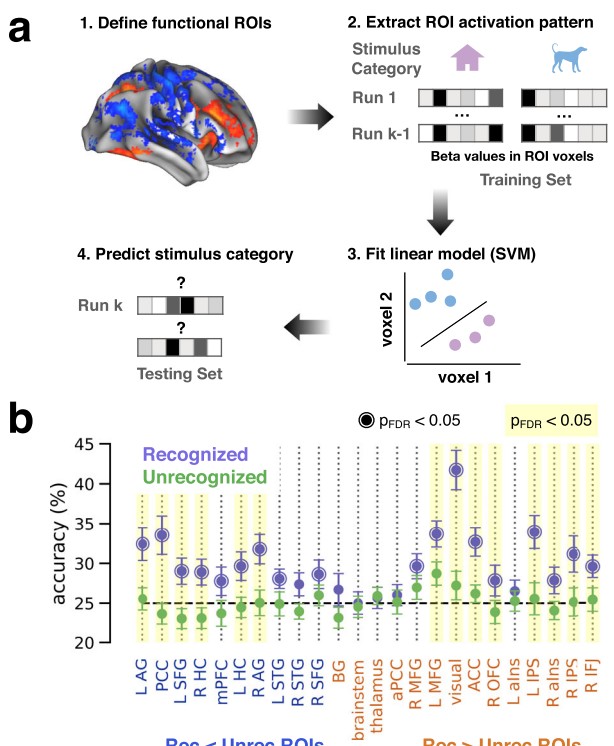

**Fig. 4 Decoding object category from multi-voxel activity patterns within ROIs. a** Schematic of the ROI-based decoding procedure. For each ROI, GLM-estimated activation values across voxels (seven voxels shown for simplicity) from a subset of the data (training data) were used to train a classifier to distinguish between stimulus categories. Held out data (testing data) were then used to test whether the classifiers could accurately decode stimulus category from new activation patterns. Separate classifiers were created for subjectively recognized objects and unrecognized objects. **b** Classifier accuracy within ROIs (locations shown in Fig. 3a) that were defined by greater activation (blue labels) or deactivation (orange labels) in response to recognized compared to unrecognized image trials. Rings around data points: significant difference from chance. Shaded columns: significant difference between recognized and unrecognized. $p < 0.05$, one-sided label permutation test, FDR-corrected across all ROIs. Error bars indicate standard error of the mean across subjects ($N = 25$).

these ROIs (defined from the recognized vs. unrecognized GLM contrast, see Methods and Fig. 3) using trials containing real object images (Fig. 4a). Stimulus category was decodable from multivoxel activity patterns in recognized, but not unrecognized, trials in most cortical areas involved in subjective recognition. Importantly, stimulus category was decodable from both activated and deactivated regions, including visual, task-positive, and DMN networks as well as bilateral hippocampi (Fig. 4b, purple circles, $p < 0.05$, label permutation test with FDR correction across all ROIs). Category decoding was unsuccessful in subcortical areas, including brainstem, basal ganglia, and thalamus, suggesting that their involvement in subjective recognition is non-content-specific. To test whether there is a difference in decoding accuracy between cortical and subcortical regions and whether it is confounded by voxel count, we fit a linear mixed model (see Methods) with parameters including: ROI location (cortical or subcortical), voxel count, their interaction, in addition to the intercept and the subject-level random effects for each parameter (model summary is reported in Supplementary Table 3, group-level data are plotted in Supplementary Fig. 2e). We observed a significant effect of ROI location (beta = 0.052 ± 0.012, estimate ± s.e.m., $p < 0.001$), indicating weaker decoding

for subcortical regions, but no significant effect of voxel count (beta = 0.011 ± 0.012, $p = 0.58$) or their interaction (beta = 0.019 ± 0.019, $p = 0.31$). This result suggests that decoding accuracy is significantly higher in cortical than subcortical ROIs, with voxel count controlled for. Notably, none of the ROIs (cortical or subcortical) had significant stimulus category decoding in unrecognized trials.

To assess the influence of subjective recognition on category information, we compared decoding accuracy between recognized and unrecognized trials. Decoding accuracy was significantly higher for recognized than unrecognized trials in almost every ROI that had above-chance decoding in recognized trials (Fig. 4b, yellow shading, $p < 0.05$, label permutation test with FDR correction across ROIs), including both activated and deactivated cortical regions. One potential concern with this analysis is that decoding performance could be affected by a bias in ROI selection, given that the ROIs were defined based on differential activation magnitudes between recognized and unrecognized trials. To address this potential issue, we performed a searchlight-based decoding analysis across the whole brain, separately for recognized and unrecognized trials. This analysis confirmed the widespread presence of object category information in recognized trials (Supplementary Fig. 2a; $p < 0.05$, cluster-based permutation test across the whole brain) and failed to detect any significant category information in unrecognized trials (Supplementary Fig. 2b), in agreement with the ROI-based results. Notably, the searchlight analysis did not reveal significant stimulus category information in brain areas that did not show BOLD response amplification, suggesting that the distributed category information is contained within those areas showing amplified activation or deactivation during successful recognition.

Because stimulus category is highly correlated with subjects' reported category in recognized trials (categorization accuracy: 79%, Fig. 1g, 'Real/Y'; also see Supplementary Fig. 1d, 'real rec'), one potential concern is that the successful stimulus-category decoding in recognized trials is in fact due to information about the reported category instead of the perceived category. If this was the case, the reported category should be decodable from unrecognized trials also. Because categorization accuracy is only at 32% in unrecognized trials (Fig. 1g, 'Real/N'; also see Supplementary Fig. 1d, 'real unrec'), this analysis provides an opportunity to dissociate stimulus- and report-related activity. We attempted to decode reported category in recognition-related ROIs and in whole-brain searchlight analysis, but neither method yielded any above-chance decoding performance. This control analysis suggests that our main analysis captures information about the perceived rather than the reported category.

Together, these results reveal a widely distributed set of brain areas encoding information about stimulus content during conscious object recognition, including both activated and deactivated cortical areas, and the hippocampus. By contrast, several subcortical areas (brainstem, basal ganglia, thalamus) show heightened activation during successful recognition with no category-specific information. Unlike the graded BOLD responses (Figs. 2–3), which have the same spatial extent in recognized and unrecognized trials but with differential response magnitudes, these results reveal an all-or-none pattern of stimulus-content-related information which exists in widespread cortical areas during conscious recognition but is undetectable in unrecognized trials. In addition, these results reveal a sharp contrast between cortical and subcortical regions: while both exhibit recognition-related amplification of responses, content-specific information was found only in cortical networks. Critically, content-specific information is not limited to activated cortical regions as previously thought, but exists saliently in deactivated cortical regions as well.

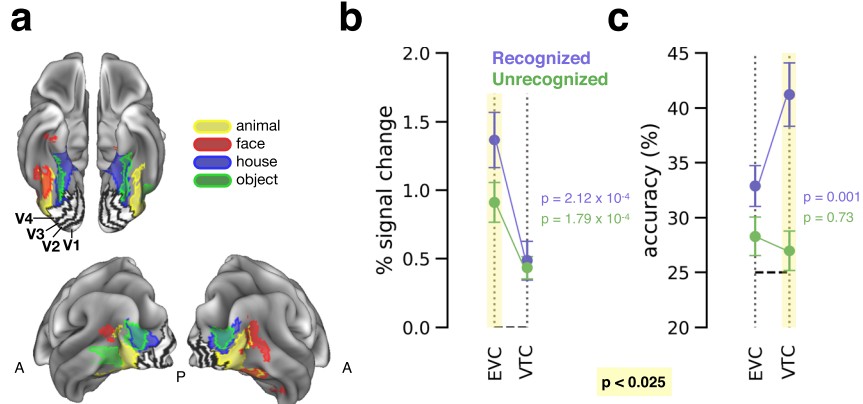

**Fig. 5 Object recognition in visual areas. a** For visualization purposes, group-level visual ROIs are plotted on a template cortical surface. Ventral temporal cortex (VTC, category-selective) ROIs were obtained from a group-level category localizer GLM analysis. Early visual cortex (EVC, V1–V4) outlines are taken from the vcAtlas[101]. Top: ventral view; bottom: posterior view. Analyses were performed using subject-level EVC and VTC ROIs derived via individual functional localizers. **b** GLM-estimated percent signal change in response to object stimuli, averaged across object categories and voxels in individually-defined visual ROIs. ROIs were averaged together within early (EVC) and higher-order (VTC) areas. Error bars indicate s.e.m. across subjects (EVC $N = 23$, VTC $N = 25$). **c** Object category decoding accuracy within individually-defined early visual cortex (EVC) and category-selective ventral temporal cortex (VTC) ROIs. Shaded columns: significant difference between recognized and unrecognized, $p < 0.025$, paired $t$-test. Error bars indicate s.e.m. across subjects ($N = 23$; EVC and VTC ROIs were matched for the number of voxels, so this analysis only included the 23 subjects with EVC ROI definition).

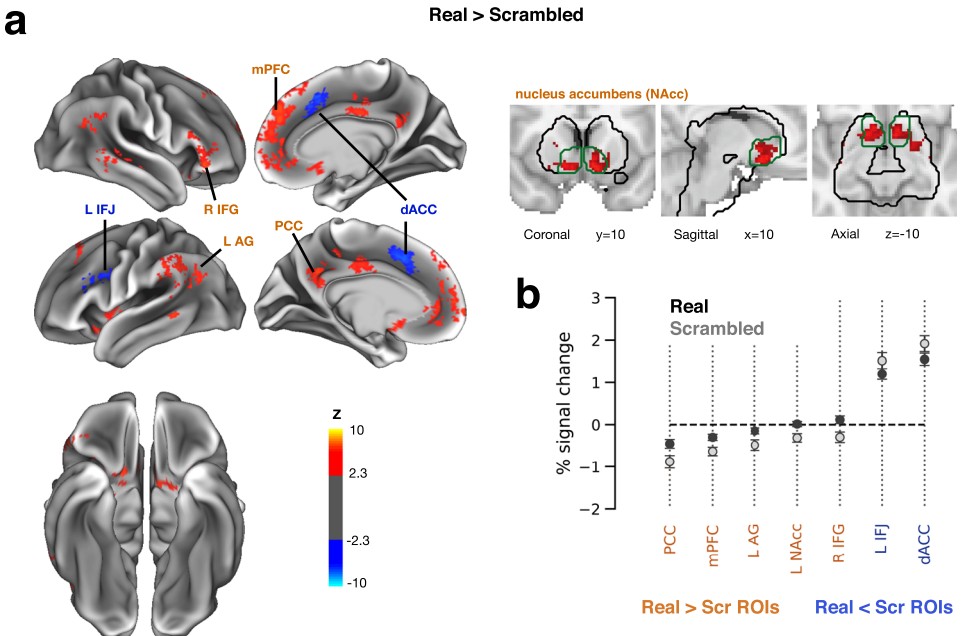

**Fig. 6 Comparing the neural responses to phase-scrambled and real object images. a** Group-level GLM contrast of real object > scrambled images. Left: data overlaid on a template cortical surface. Right: data presented in subcortical regions only (within black contour; green contour indicates nucleus accumbens), plotted on MNI152 template brain volume. Data were thresholded using cluster-based FWE-correction in FSL, with a voxel-defining Z statistic threshold of $p < 0.01$ and a cluster size threshold of $p < 0.05$. **b** GLM-estimated percent signal change of stimulus-triggered responses, averaged across voxels within clusters shown in **a**. Orange and blue labels denote clusters with positive and negative contrast values, respectively. In all plots, data are averaged across object categories and error bars indicate s.e.m across subjects ($N = 25$). AG: angular gyrus, PCC: posterior cingulate cortex, mPFC: medial prefrontal cortex, IFG: inferior frontal gyrus, acc: accumbens, IFJ: inferior frontal junction, dACC: dorsal anterior cingulate cortex, L: left, R: right.

**Subjective object recognition in early and object-selective visual cortex.** Previous studies on conscious object recognition using masking paradigms found increased activity and category selectivity with recognition in category-selective visual areas but not early visual cortex[34–36,45]. To allow a direct comparison of our data with these findings, we identified early visual cortex (EVC, including V1–V4) and category-selective ventral temporal cortex (VTC) ROIs for each subject from independent functional localizers (see Methods, *Visual object localizer* and *Retinotopic*

*mapping*; for approximate ROI locations, see Fig. 5a). The VTC ROIs were defined by contrasting localizer responses to each object category against the three other categories to derive category-selective ROIs.

EVC exhibited greater activation in response to recognized than unrecognized object images (Fig. 5b, $t_{22} = 3.289$, $p = 0.0033$, paired $t$-test), consistent with the whole-brain analysis (Fig. 3a). We found this effect in most sub-regions of EVC (Supplementary Fig. 3a, left: V1: $t_{22} = 3.97$, $p = 6.47 \times 10^{-4}$; V2: $t_{22} = 2.62$,

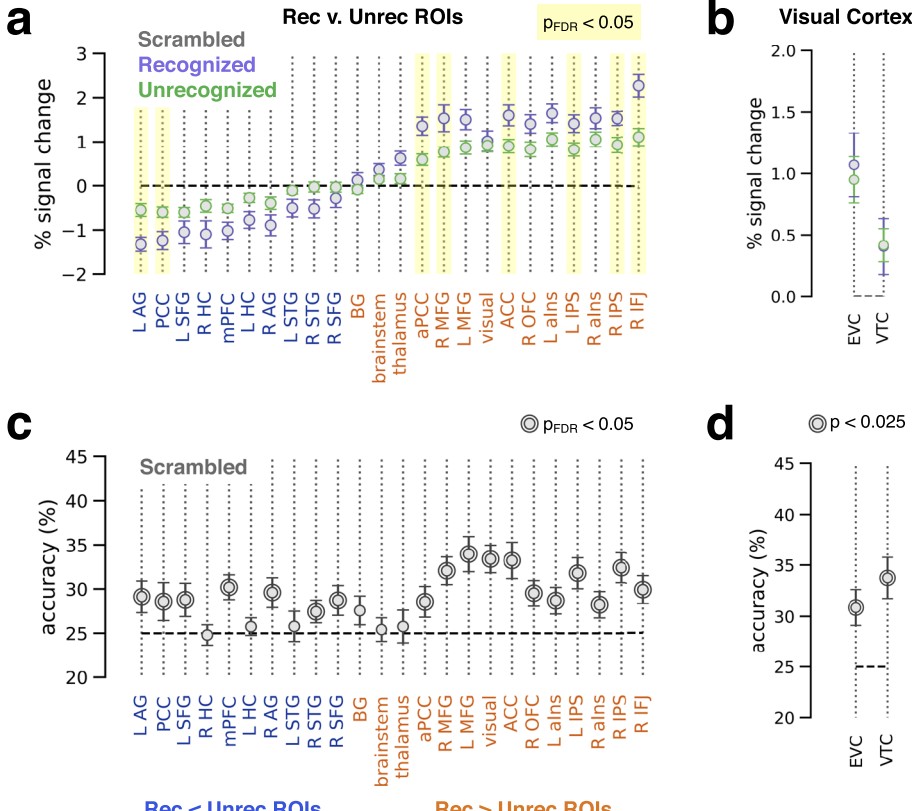

**Fig. 7 Neural responses and category information elicited by phase-scrambled images. a** GLM-estimated percent signal change of stimulus-triggered response for each ROI defined in Fig. 3. Data are separated by subjective recognition report, averaged across stimulus categories. Shaded columns: significant difference between recognized and unrecognized, paired $t$-test, $p < 0.05$, FDR-corrected across ROIs ($N = 25$). **b** Same as **a**, but for individually-defined early (EVC, $N = 23$) and category-selective (VTC, $N = 25$) visual areas. **c, d** Stimulus category decoding accuracy within recognition-contrast ROIs (**c**, same ROIs as in **a**, $N = 25$) and individually-defined visual ROIs (**d**, $N = 23$). Rings around data points: significant difference from chance, one-sided label permutation test, $p < 0.05$ FDR-corrected across ROIs (**c**) or $p < 0.025$ (**d**). In all plots, error bars indicate s.e.m. across subjects.

$p = 0.016$; V3: $t_{22} = 1.42$, $p = 0.17$; V4: $t_{21} = 2.51$, $p = 0.02$), including the primary visual cortex which did not show an object recognition effect in previous studies using visual masking[34,36,45]. Two possible explanations for this discrepancy are if masking interrupts recognition-related feedback to V1, or if masks elicit strong neural responses that obscure weaker recognition effects. Interestingly, activation magnitude in VTC did not significantly vary with recognition (Fig. 5b, $t_{24} = 0.44$, $p = 0.66$, paired $t$-test; individual sub-regions results shown in Supplementary Fig. 3a, right), although the whole-brain analysis did reveal a significant recognition effect in parts of the ventral temporal cortex (Fig. 3a). The overall BOLD responses were weaker in VTC than in EVC (Fig. 5b, recognized: $t_{22} = 4.43$, $p = 2.1 \times 10^{-4}$; unrecognized: $t_{22} = 4.50$, $p = 1.8 \times 10^{-4}$; paired $t$-test). This pattern of results may be due to the VTC ROIs being defined by category selectivity (instead of intact vs. scrambled objects) and, as a result, producing relatively weak responses to non-preferred categories.

We next tested whether object category information, decoded from multivoxel activity patterns, varied with subjective recognition. For this analysis the voxel count was equated between EVC and VTC regions (see Methods, *MVPA*). Category decoding was significantly better than chance in both areas for recognized trials (Fig. 5c; EVC: $p = 0.001$; VTC: $p = 0.001$; label permutation test; for results from individual sub-regions see Supplementary Fig. 3b), and lower for unrecognized trials (EVC: $p = 0.018$; VTC: $p = 0.065$; label permutation test). Further, decoding performance was significantly better for recognized than unrecognized trials in VTC (difference of $14.2 \pm 2.5\%$, $p = 0.001$, label permutation test),

and a smaller difference was found in EVC (difference of $4.6 \pm 2.1\%$, $p = 0.028$, label permutation test). Lastly, decoding accuracy was higher for VTC than EVC for recognized objects ($p = 0.001$, label permutation test) but not unrecognized objects ($p = 0.73$, label permutation test).

These results are consistent with prior studies pointing to the critical involvement of high-level visual cortex in object recognition[8,45], but also reveal differential activity in early visual cortex that correlates with conscious object recognition.

**Scrambled images elicit heightened processing in prefrontal and default-mode areas.** We next investigated the differences in neural processing between real object images and scrambled images. Scrambled images elicited a substantial but lower fraction of recognition reports (28%) as compared to real object images (48%, Fig. 1f), and a similar but poorer pattern of category discrimination accuracy (Fig. 1g). Previous studies showed that scrambled images recruit category-selective visual cortex to a much lesser extent than real object images[45]; as such, scrambled images are often used as the baseline condition in functional localizers to define higher-order visual areas. Thus, one might expect that scrambled images elicit weaker higher-order visual cortical activity and a failure of propagation of category information to frontoparietal areas.

We first compared the amplitude of neural responses elicited by real object and scrambled images (Fig. 6a, cluster sizes and MNI coordinates are reported in Supplementary Table 4). Interestingly, no difference is found between real object and

scrambled images in visual cortex. This lack of difference is likely due to image presentation at a liminal contrast which introduces a high level of sensory uncertainty, unlike object localizers that use high-contrast images.

Instead, scrambled images elicited stronger activation than real object images in the left inferior frontal junction (IFJ) and bilateral dorsal anterior cingulate cortex (dACC) (Fig. 6a, b, blue). Both clusters are contained within larger clusters observed in the recognized > unrecognized contrast using real object images (compare to Fig. 3a). Surprisingly, no brain area showed stronger activation to real objects compared to scrambled images. Instead, a positive real > scrambled contrast (Fig. 6a, red) resulted from scrambled images eliciting stronger deactivation in DMN regions (including mPFC, L AG and PCC), right inferior frontal gyrus and part of the basal ganglia (nucleus accumbens) ($p < 0.05$, FWE-corrected; Fig. 6a, b, red-orange). The DMN regions overlapped with clusters revealed by the unrecognized > recognized contrast using real object images (compare to Fig. 3a).

Importantly, this difference in activation patterns between real object and scrambled images cannot be attributed to a lower recognition rate for scrambled images. Both IFJ and dACC show stronger activation in recognized than unrecognized trials (for both real and scrambled images, Figs. 3a and S4), as well as stronger activation to scrambled images. Similarly, DMN regions show stronger deactivation in recognized trials, as well as stronger deactivation to scrambled images. By contrast, we suggest that the enhanced activation of IFJ and dACC and deactivation of DMN regions to scrambled images likely reflect inferential processing when stimulus input is highly ambiguous. Several additional analyses described below further support this idea.

First, using the ROIs identified from the recognition contrast of real object images (Fig. 3a), we extracted their responses to scrambled images, separated by recognition status (Fig. 7a). This analysis revealed a similar pattern of activation and deactivation to real object and scrambled images (compare Fig. 7a with Fig. 3b), with comparable magnitudes of the recognized—unrecognized difference (Supplementary Fig. 4b). The whole-brain GLM contrast of recognized vs. unrecognized trials also appears qualitatively similar albeit weaker for scrambled images (compare Supplementary Fig. 4 to Fig. 3a). This pattern of results, showing similar recognition-related effects between real and scrambled images, is further evidence for a genuine 'false perception' interpretation of "recognized" scrambled images.

Second, using the same ROIs, we attempted to decode the category of scrambled images (i.e., category of the original object image from which the scrambled image was generated, whose low-level image features are preserved by phase-scrambling). Some of these ROIs overlap with the ROIs obtained from the real vs. scrambled image contrast (Fig. 6a), allowing us to test whether areas whose responses differ between real and scrambled images contain content-specific activity. We did not take subjective recognition status of scrambled images into account here, because low trial counts for recognized scrambled images precluded effective cross-validation within that subgroup. The category of scrambled images could be decoded from nearly all cortical ROIs investigated (with the exception of left superior temporal gyrus), but no subcortical ROIs (Fig. 7c; $p < 0.05$, label permutation test with FDR correction across ROIs). The cortical areas with heightened activation (L IFJ and dACC, as components of L MFG and ACC) or deactivation (DMN areas) in scrambled image trials (Fig. 6) all contained decodable category information. This result further strengthens the evidence for heightened processing of stimulus/perceptual content in these areas when scrambled images are presented.

Within visual areas, category decoding of scrambled images was significantly above chance in both EVC ($p = 0.001$, label permutation test) and VTC ($p = 0.001$) (Fig. 7d). Subjective recognition of scrambled images did not correlate with response magnitude in either EVC or VTC (Fig. 7b).

Importantly, stimulus category was decodable across the brain in scrambled image trials but not in unrecognized object image trials (Fig. 4b), despite the fact that the latter but not the former contains a meaningful object stimulus. Together, these results suggest that scrambled images, which contain low-level visual features but lack object information, trigger a different mode of neural processing that is more reliant on frontoparietal areas (including dACC, IFJ, and DMN areas). They support the idea that higher-order processes are recruited to infer stimulus category under uncertainty, which sometimes triggers a pattern of large-scale brain activity similar to that elicited by recognition of real meaningful objects and as a result drives false perception—seeing a meaningful pattern out of a meaningless noise pattern.

## Discussion

Here we report large-scale brain signatures of conscious object recognition. Contrary to leading theories of conscious perception, we observed widespread brain responses to both recognized and unrecognized object images with nearly identical spatial extent but different response magnitudes. Outside visual cortex, these responses encompassed a 'task-positive' network including cortical and subcortical regions, which showed stronger activation to recognized than unrecognized object images, and the default-mode network, which showed stronger deactivation to recognized images. Strikingly, nearly all of the responsive cortical regions, either activated or deactivated, exhibited activity patterns containing information about the perceptual content, while content-specific activity was absent in subcortical regions. Lastly, phase-scrambled images, which are devoid of any meaningful content but preserve low-level visual features, triggered false perception accompanied by heightened processing in specific prefrontal and DMN regions, suggesting that these regions may be involved in top-down inferential processing when sensory input has a high level of uncertainty.

Stimulus category could be decoded from nearly all of the activated as well as deactivated cortical regions in recognized trials, but not when recognition failed (Fig. 4b). We also observed widespread object category information in a localizer task using high-contrast images (Supplementary Fig. 2d), suggesting that the large-scale broadcasting of content-specific information in our main task is not due to potentially increased task difficulty. In addition, a control analysis showed that this result likely reflects perceived rather than reported category, since reported category was undecodable in unrecognized trials that had a similar trial count. In contrast to recurrent-processing theory (RPT) and integrated information theory (IIT) proposals suggesting that the content-specific neural correlates of conscious perception are localized to posterior sensory areas, our results instead reveal that subjective recognition triggers content-specific activity across widely distributed cortical networks.

Our results during subjective recognition are broadly consistent with the global neuronal workspace (GNW) theory, which predicts that conscious contents are broadcast throughout a distributed brain network. Evidence for perceptual content representation in this distributed network during conscious but not unconscious processing was previously weak[13,52]; our findings substantially strengthen the evidence for global broadcasting as a correlate of conscious perception. Specifically, we show that neural representations of content exist during subjective recognition in PFC, ACC, PCC—brain areas proposed by the GNW theory as hubs for information broadcasting during conscious processing[5].

However, our results are inconsistent with the GNW theory's formulation of unconscious processing. The GNW theory suggests that unconscious processing involves a weak, slow-decaying wave of activation along feedforward pathways, potentially reaching prefrontal cortex, while only conscious processing triggers global activation ('ignition') involving a more extensive network of reverberant loops, which enables access of a given piece of information to a wider array of brain regions—especially frontoparietal areas[5,47,53]. While our results show non-content-specific activation as a correlate of unconscious processing of objects, it is difficult to explain the wide spatial extent of these responses, nearly identical to that of the responses to recognized images, with a purely feedforward propagation thesis. This result calls into question the previous focus on the spatial extent of brain responses as a defining correlate of conscious perception, and raises instead the importance of content-specificity of such activity. That is, responses to recognized images are both stronger and content-specific (and critically, in both activated and deactivated networks, a point we elaborate on below).

One might wonder whether our result of widespread overlapping responses in recognized and unrecognized trials partially stems from the detection of low-level stimulus features, which could potentially be consciously perceived in unrecognized trials. If this were the case, one would expect to observe above-chance category decoding accuracy in unrecognized trials because low-level features of object stimuli differ across object categories[38,43]. Our data and experimental procedure are sensitive to such category-specific low-level features of object images, as is evident from successful category decoding from brain responses to scrambled images (Fig. 7c). Because the responses to unrecognized real images did not contain any information about object category (Fig. 4b), it is unlikely that the widespread activation stems from conscious perception of low-level image features. Instead, our finding of broad cortical activation and deactivation in unrecognized trials is consistent with increasing evidence showing that unconscious processing can penetrate into higher-order association cortices including the prefrontal cortex[12,54,55]. Importantly, our results substantially extend this prior literature by showing that the spatial extent of cortical activation can be similar between success and failure of subjective recognition, while the response amplitudes have a graded modulation (high vs. low) and information content has an all-or-none pattern (present vs. absent) across this widespread cortical network.

The default-mode network (DMN) is not part of the postulated neural mechanisms underlying conscious perception by any of the current prominent theories[3–5,56], although an earlier hypothesis proposed that the automatic associations served by this network facilitate perception and cognition[57]. We observed that the DMN deactivates to both unrecognized and recognized object images, and this deactivation is stronger for recognized images (Figs. 2 and 3). This finding is consistent with a wealth of previous studies showing DMN deactivation in externally oriented tasks[20]. Given DMN's known involvement in self-referential processing[58], one might propose that the stronger deactivation observed in recognized trials is due to a reduction in task-irrelevant thoughts when recognition is successful. However, this is unlikely because during successful recognition, despite a stronger deactivation, stimulus content information is encoded throughout the DMN (9 out of 10 DMN regions had significant decoding, Fig. 4b) while stimulus content is not encoded when recognition was unsuccessful. Thus, contrary to being task-irrelevant, the strong deactivation in DMN accompanying subjective recognition actively encodes perceived stimulus content.

This pattern of findings is consistent with recent observations of content-specific deactivation of DMN. For example, the lateral parietal component of the DMN (angular gyrus) exhibits stronger

deactivation and increased pattern-based information during successful memory encoding[59], similar to the dissociation between activity level and information strength observed herein. Another recent study revealed retinotopically organized maps in the DMN whereby the DMN selectively deactivates depending on the spatial location of a visual stimulus[21]. An earlier study from our laboratory[22] showed that the DMN contains content-specific information during prior-guided recognition of degraded images. In that study, recognition was only possible after relevant prior knowledge has been acquired, and prior-guided recognition was associated with reduced deactivation as well as enhanced content-specific information. Taken together with our current findings, these results suggest that the effect of recognition on the gross activity level in the DMN depends on whether recognition is driven by prior knowledge[22] (whereby deactivation is reduced) or fluctuations in spontaneous brain activity as in the present paradigm (whereby deactivation is stronger). However, in both cases, pattern-based information increases with recognition.

The hippocampus is functionally coupled with the DMN[46] and we observed a similar pattern of stronger deactivation with enhanced category information during subjective recognition in the hippocampus (Figs. 3b and 4b). The hippocampus has an established role in memory and relational processing, but existing studies largely suggest that its involvement in conscious perception is limited to scenes rather than objects[60,61]. However, consistent with our findings, a previous intracranial study found that event-related potentials directly recorded from the hippocampus distinguished between identified and unidentified object stimuli[62].

Together, these findings provide insight into the role of DMN in subjective object recognition, and urge future theories on conscious perception to take this network, a hub of corticocortical communication[17,18], into consideration.

Several subcortical regions, including the brainstem, thalamus and basal ganglia, showed a significant recognized > unrecognized activity contrast (Fig. 3) but did not contain any detectable category information (Fig. 4b). This result was not due to any systematic difference in ROI sizes between cortical and subcortical regions (Supplementary Table 1, Supplementary Fig. 2e). It is possible that this null result is partially due to lower fMRI sensitivity to subcortical activity. Currently, no non-invasive neuroimaging method in humans provides equal sensitivity to cortical and subcortical regions. However, prior work demonstrates that 7 T fMRI can measure subcortical signals involved in perceptual processes with a considerable improvement over 3 T fMRI[63]. It remains possible that subcortical regions encode content-specific information in finer-grained activity patterns within a single voxel, which we cannot detect. However, our finding suggests that content-specific information in these subcortical regions is at least much weaker than that in the cortical networks, resulting in undetectable content information in subcortical regions using the current methods involving high-field fMRI, in contrast to the strong content-specific information in widespread cortical networks. Below we discuss the implication of this finding.

In the thalamus, recognition-related activity centers around the mediodorsal (MD) nucleus. The MD-PFC circuit has been implicated in attentional control and cognitive flexibility[64], and a recent rodent study showed that MD sends a content-invariant signal to PFC which amplifies and sustains rule-specific neural sequences in the PFC[65]. The thalamus' role in our task may thus be to amplify object information in the cortex for it to cross the threshold of recognition, or to ensure that recognized object information is maintained in cortex for the duration of the trial.

Previous studies have shown that the basal ganglia are involved in perceptual decision-making[23]. Specifically, the dorsal striatum

implements adjustments in decision bias[66]; as such, the stronger response in recognized trials in the caudate and putamen may indicate such a biasing effect. Stimulus-triggered responses in nucleus accumbens are often linked to reward anticipation and its associated sensation of pleasure[67]. While decisions made in our task were not externally rewarded, successful recognition can be genuinely rewarding, which might contribute to the elevated response in nucleus accumbens in recognized compared to unrecognized trials.

Several brainstem nuclei also showed higher activity during subjective recognition. These nuclei, including the pedunculo-pontine nucleus[68], locus coeruleus[69], ventral tegmental area[70], and dorsal raphe nucleus[71], all modulate arousal. Arousal-related activity in the brainstem neuromodulatory system correlates with stimulus detection[26] and the precision of visual cortical representations[72]. Hence, these brainstem nuclei might contribute to subjective recognition either via generally supporting sensory detection or by refining the representation of high-level object information in cortex.

These results thus reveal a previously underappreciated aspect of the neural machinery supporting conscious recognition: content-invariant activity in several subcortical circuits that may either enhance or enable content-specific activity in widespread cortical networks underlying subjective object recognition. Although the role of subcortical brain regions in regulating arousal and states of consciousness is well established, their involvement in conscious perception remains unclear[5,32]. Our results provide initial findings that both indicate their involvement in conscious perception and outline an important difference between subcortical and cortical mechanisms.

Surprisingly, we were able to decode category information from scrambled-image trials across the brain, in both activated and deactivated brain regions (Fig. 7c). The scrambled images do not contain objects, but only low-level image features which differ across categories and are preserved by the phase-scrambling procedure. The successful category decoding in scrambled images, however, cannot be attributed to the presence of low-level image features alone, because we were unable to decode stimulus category in unrecognized real object image trials (Fig. 4b), despite these having higher trial counts (scrambled images account for 16.7% of all trials; unrecognized real object image trials account for 43.3%). Instead, we suggest that in our paradigm, scrambled images elicit inferential processes attempting to resolve stimulus content under high sensory uncertainty, which sometimes result in meaningful percepts ('false perception'). In this process, stimulus information may be propagated across the brain in an attempt to resolve its content and can thus be decoded from neural activity.

Studies of visual object recognition typically find a strong and replicable result of reduced activation in ventral temporal visual cortex when viewing high-contrast versions of scrambled images as compared to real object images[45]. Contrary to our initial prediction based on these prior studies, we did not detect this effect in our data (Fig. 6a). This null finding is likely related to the liminal nature of our stimuli: in our study, the contrasts of real object images were titrated to an individual's perceptual threshold, and scrambled images were presented at the same contrast. Thus, the brain may not receive enough information to conclude that a scrambled image is actually lacking meaningful content compared to liminal object images. Instead, when no real object information is present in bottom-up sensory input, top-down inferential processes may be especially active, as supported by our finding that scrambled images elicit stronger activation in the dACC and IFJ than real object images (Fig. 6, blue). The IFJ is involved in controlling object-based attention[73], and the dACC is associated with uncertainty resolution[74] and hypothesis testing[75].

In addition, recent rodent studies suggest that the dACC sends predictions about visual contents back to visual cortex via feedback connections[76,77]. Scrambled images, lacking object-level content to be eventually detected, may elicit more extensive hypothesis testing than images containing real objects. This process may be facilitated by connections between IFJ and dACC within the cognitive control network[78].

Similar results were found in DMN regions: scrambled images elicited stronger deactivation than real object images. As with IFJ and dACC, these DMN regions contain content-specific information about the category from which the scrambled images were generated (Fig. 7c), which strongly correlates with the perceived category during false perception (i.e., false-alarm trials; see Fig. 1g, 'Scram/Y'). Thus, the amplified deactivation to scrambled images in DMN does not reflect suppression of task-irrelevant activity; instead, the most parsimonious explanation is that the DMN regions are also involved in top-down inferential processing and represent the content of perceptual outcome in their deactivation. This is an exciting hypothesis for future studies to address.

Is the neural mechanism for subjective recognition the same for scrambled stimuli as for real object stimuli? It is still possible that consciously perceived content is encoded differently between real and scrambled stimuli, but our study was not designed to answer this question (the number of scrambled image trials reported as recognized was too low to permit content decoding in this group of trials). However, the recognized > unrecognized GLM contrasts suggest substantial overlap between real and scrambled images with nearly identical contrast magnitudes (Supplementary Fig. 4b; also compare Fig. 3b to Fig. 7a). This is consistent with the notion that subjective recognition of real and scrambled object images both involve perceptual inference, but inferential processes are engaged to a higher degree under greater sensory uncertainty[79]. Supporting this idea is our finding that the same regions showing heightened responses to scrambled images, dACC and IFJ (activation) and DMN (deactivation), also correlated with subjective recognition in both real and scrambled image trials.

There are several limitations to the present study. First, to probe subjective recognition, we relied on subject's report — the gold-standard approach. Several recent studies have developed no-report paradigms in the context of binocular rivalry and inattentional blindness to investigate the neural correlates of conscious perception[80]. Thus far, studies using this approach have reported mixed findings, with some reporting strongly reduced frontoparietal activation[81] and others reporting preserved PFC encoding of conscious content[82]. It is unclear if a no-report paradigm for genuine and false object recognition, paralleling the current paradigm, could ever be established. But if so, such a paradigm could be used to shed light on this contentious issue and help reveal the relationship between neural mechanisms involved in subjective recognition and the (automatic or voluntary) reporting of this recognition. Second, using 7 T fMRI, we recorded whole-brain activity with fine spatial resolution and high sensitivity to cortical and subcortical regions. However, due to the slow hemodynamic response, this technique cannot capture fast neural dynamics. We cannot, for example, address whether unrecognized trials contain evidence for the GNW notion of "failed ignition"[14], in which transient and weak content-specific activation fails to ignite the global workspace[5]. Future investigation using intracranial recordings or an MEG-fMRI data fusion approach[83,84] will be able to fill in the temporal dimension. Third, following prior studies of conscious object recognition, the "recognized" versus "unrecognized" responses are influenced by individual participants' criteria for a meaningful object percept. While such cross-subject variability is not the focus of the present

study, it is of interest for future work to understand individual differences in object perception. Lastly, in this study, we focus on activation magnitude and information content of stimulus-triggered neural responses that underlie genuine and false object recognition. Previous studies e.g.,[12] including our own[38,85] have shown that spontaneous brain activity fluctuations in the pre-stimulus period influences conscious perception and subjective recognition. An important direction for future studies is to better understand the relationship between pre-stimulus brain state and stimulus-triggered activity, which can exhibit complex, nonlinear interactions[86,87].

In sum, our study demonstrates that conscious object recognition is supported by a much more extensive brain network than was previously known. Rather than recruiting additional cortical areas, subjective recognition is correlated with amplified positive and negative responses across visual, frontoparietal, subcortical and default-mode networks. At the same time, stimulus content information, undetectable during unrecognized trials, becomes widely distributed across activated and deactivated cortical regions during successful recognition. By contrast, subcortical regions exhibit enhanced activation during subjective recognition that is non-content-specific, suggesting that they may facilitate content-specific activity in the cortex during conscious perception. Our results further highlight the dorsal anterior cingulate cortex, inferior frontal junction, and the default-mode network as hubs for perceptual inference, contributing to both genuine and false recognition. These results point to a more comprehensive framework for conceptualizing neural mechanisms underlying conscious object recognition, which involve content-specific activity across activated and deactivated cortical networks and content-invariant subcortical activity.

## Methods

**Participants**. All participants ($N = 38$, 26 females, mean age 27.18, range 20 to 38) provided written informed consent. The experiment was approved by the Institutional Review Board of New York University School of Medicine (protocol #15-01323). The participants were right-handed, neurologically healthy, and had normal or corrected-to-normal (wearing contact lenses) vision. 10 enrolled participants did not complete the experiment due to the following reasons: poor performance in image contrast staircasing task (4 subjects, see below), request to end experiment early (6 subjects). Data from 3 subjects who completed the experiment were not analyzed due to poor performance in the main object recognition task (see below). Data from a total of 25 subjects were used in the final analysis.

**Experimental procedure**. The experiment consisted of two separate scanning sessions, conducted on two separate days (Fig. 1a). Day 1 included image contrast staircase (with simultaneous anatomical MRI acquisition) and functional localizers. Day 2 began with an 8-minute recognition task to confirm the threshold contrasts obtained on day 1, and if necessary, an additional 8-minute staircase session to adjust for threshold changes across experiment days. This was followed by the main object recognition task.

**Experimental stimuli**. Stimulus images were created as described in our previous work[38]. Briefly, images from four categories: faces, animals, houses, and objects (Fig. 1c) were selected from public domain labeled photographs or from Psychological Image Collection at Stirling (PICS, http://pics.psych.stir.ac.uk/), resized to 300×300 pixels and converted to grayscale. The actual experimental stimuli from PICS are not available for commercial use; therefore, the images shown in Figures were downloaded from https://www.pexels.com/ and are presented to demonstrate the outcome of the image processing procedure described below. Pixel intensities were normalized by subtracting the mean and dividing by the standard deviation. Images were then filtered with a 2D Gaussian smoothing kernel with a standard deviation of 1.5 pixels and 7 × 7 pixel size (imgaussfilt, MATLAB). Five unique images were included in each category, resulting in 20 unique real images in total. Scrambled images were created by randomizing the 2D Fourier transform phase of one image from each category. The edges of all images were gradually brought back to background intensity by multiplying the image intensity with a Gaussian window (standard deviation 0.2). Stimuli were presented using the Psychophysics Toolbox[88] in MATLAB via an MRI-compatible LCD monitor (BOLDScreen, Cambridge Research Systems) with a 120-Hz refresh rate, located 210 cm behind the center of the scanner bore. Subjects viewed the screen using an eye mirror

attached to the head coil. Stimuli subtended 4.3 degrees of visual angle at central fixation.

**Image contrast staircase**. On Day 1, subjects performed an adaptive staircase procedure "QUEST"[89] in the scanner in order to determine individual image contrast ($c$) yielding a recognition rate of 50% (proportion of "yes" responses to the second question). The image pixel intensity, $I$, at a given contrast, $c$, was calculated as:

$$I(c) = b(I_{scaled} * c + 1) \tag{1}$$

where $b$ is the background intensity (set to a constant value of 127) and scaled pixel intensities ($I_{scaled}$) were obtained by rescaling the image pixel intensities between $-1$ and 1. As a result, the lightest pixel value in the image was equal to $I_{max} = b(1 + c)$ and the darkest $I_{min} = b(1 - c)$. Therefore, we defined the contrast of a presented image as:

$$c = \frac{I_{max} - I_{min}}{2b} \tag{2}$$

which ranged between 0 and 1.

Subjects performed the QUEST procedure in the scanner under conditions similar to the main task performed on Day 2, except timing parameters were shorter. The inter-trial interval (ITI) was randomly selected from 0.75 or 1 s, and the fixation period between stimulus presentation and categorization prompt was 0.5 s. Because the shorter ITI allowed subjects to better predict the time of stimulus onset than in the main task, effective 50% threshold contrasts were lower in QUEST than in the main task. The target threshold in QUEST was thus set to 55% rather than 50% to compensate. The threshold contrast for each image was identified using an independent QUEST process containing 40 trials. 20 such processes (one for each real image) were executed such that different images were interwoven into a total session of 800 trials. The task was broken into four blocks of 200 trials each, and subjects were allowed to rest in between blocks. Task performance was considered acceptable if categorization accuracy for recognized images was at least 30% higher than for unrecognized images, and if the QUEST procedure successfully converged on 55% recognition rate for each image. On Day 2, we first tested whether the threshold contrasts still yielded 50% subjective recognition. Subjects completed 80 trials (4 trials per image) with the following timing parameters: ITI of randomly 2 or 3 s, post-stimulus fixation of 2 s. If subjects reported to recognize more than 80% or less than 30% of trials, they performed a shorter QUEST task with the same timing parameters to account for threshold changes across days. These 14 subjects completed two interwoven QUEST processes with 40 trials each, including all real images. Instead of adjusting the contrast of each image individually, the staircasing target was 50% recognition across all images. We thus accounted for any change in overall recognition threshold across days, but not for any new differences between images.

**Main object recognition task**. Subjects performed the main task in fifteen 24-trial runs lasting ~7.5 min each on Day 2 (see Fig. 1a), for a total of 360 trials. Two subjects did not complete all 15 runs but their data were still included (see Supplementary Table 5), and one subject performed 3 extra runs for a total of 18. Each run contained one trial per stimulus (20 real object, 4 scrambled), and subjects were not informed of the presence of scrambled images. Subjects were allowed to rest between runs. The entire task lasted about 2.5 h. Each trial (Fig. 1d) began with a fixation cross on a gray background for a period of random duration between 6 and 20 s, with the set of fixation durations following an exponential distribution. The timing jitter ensured that the subject could not predict the onset of the stimulus. The stimulus image was then presented behind the fixation cross for 8 frames (66.7 ms). Image intensity increased gradually from 0.01 (1st frame) to threshold intensity (8th frame). After the stimulus disappeared, the fixation cross persisted for another randomly chosen duration of either 4 or 6 sec (following an exponential distribution). The luminance of the blank screens before and after the stimulus was equal to the background luminance of the stimulus screen. Each trial ended with 2 sequential questions about the stimulus, each appearing for 2 s. The first question asked what category the image belonged to: animal, face, house, or object. In case the object was not detected, subjects were instructed to make a genuine guess. This question targeted objective performance (i.e., accuracy). The response button mapping for this question was randomized across trials. The second question targeted subjective experience and asked whether subjects had a "meaningful visual experience" of the object stimulus ("yes" or "no"). Meaningful was defined during a practice session outside of the scanner as something that makes sense in the real world, as opposed to an image of pure noise or meaningless shapes. Subjects were instructed to respond 'yes' if they had any sense of the object presented, regardless of whether the particular content was identified. For example, perceiving a furry tail may constitute a meaningful experience of an animal, though it could belong to either a monkey or a cat. Subjects indicated their answers to the questions via a fiber-optic button box attached to their right hand (Psychology Software Tools, Inc.). Task performance was considered acceptable if recognition rate for each category was above 15%, and if categorization accuracy for recognized images was at least 30% higher than for unrecognized images.

**MRI data acquisition**. Experiments were conducted in a Siemens 7 T MRI scanner using a 32-channel head coil with internal head cage at New York University Center for Biomedical Imaging. High-resolution (1.0 mm isotropic voxels) T1-weighted MPRAGE images were acquired with the following parameters: FOV 256 mm, 192 sagittal slices, TR 3000 ms, TE 4.49 ms, flip angle 6°, fat suppression on. For intensity normalization, proton density (PD) images were acquired with the following parameters: FOV 256 mm, 192 sagittal slices, 1.0 mm isotropic voxels, TR 1760 ms, TE 2.57 ms, flip angle 6°, bandwidth 280 Hz/Px. Blood oxygen level-dependent (BOLD) fMRI images were acquired with the following parameters: FOV 192 mm, 54 oblique slices covering all of cortex, voxel size 2.0 mm × 2.0 mm, slice thickness 2.0 mm with distance factor 10%, TR 2000 ms, TE 25 ms, multiband factor 2, GRAPPA acceleration 2, phase encoding direction P- > A, flip angle 50°.

**Anatomical MRI data preprocessing**. Preprocessing of anatomical images was performed using FSL[90] version 5.0.10. To extract anatomical brain images, T1 and PD images were first skull-stripped using BET[90]. The resulting PD brain image was smoothed using a 2 mm kernel, and PD voxels with values less than 1 (an arbitrary threshold, considered to be noise after visual inspection of several example images) were set to 0. The T1 brain was divided by the smoothed PD brain to correct for inhomogeneities[91]. The resulting T1/PD image was finally masked by the smooth PD brain, to remove remaining noise around the edges of the brain. In preparation for cortical surface reconstruction with Freesurfer's recon-all command (http://surfer.nmr.mgh.harvard.edu), a new full-head T1 image was created by taking the original T1-weighted image including skull, and replacing the inhomogeneous brain with the new T1/PD brain.

**Functional MRI data preprocessing**. We performed the following pre-processing steps for each task run using FSL's FEAT tool. We first corrected for motion artifacts using MCFLIRT, which aligns each volume to the volume acquired at the middle of the run and estimates 3 dimensions of head rotation and translation across time (6 DOF). We removed one or more blocks from 3 subjects due to excessive motion defined as a >6 mm spike within the relative mean displacement timecourse (Supplementary Table 5). To account for the long whole-brain acquisition time (2000 ms), we interpolated signal from each slice to the middle of each TR (slice timing correction). We then extracted the brain using BET, applied spatial smoothing (3.0 mm FWHM for main task, 4.0 mm FWHM for category localizer), applied high pass filtering with a temporal cutoff of 150 s to remove slow drifts, and applied grand mean scaling to normalize the mean voxel intensity across runs and scanning sessions. The functional images were registered to anatomical images using linear boundary-based registration (BBR[92]) and anatomical images were registered to a standard brain image (MNI 152) using linear registration (12 DOF). Artifacts related to motion, arteries or CSF pulsation were removed using ICA (MELODIC) and inspecting 30-40 components that together explain ~75% of variance in the BOLD signal[93].

**General linear model (GLM) analysis**. We used a GLM (implemented with FEAT tool in FSL) to assess stimulus-triggered activation. For each task run, we created one regressor for each possible combination of stimulus category (face, animal, house, object), recognition report (yes, no), and image type (real, scrambled), all aligned to stimulus onset. Thus, each run had up to 16 stimulus regressors, given 4 categories × 2 recognition statuses × 2 image types. Two additional control regressors modeled 1) trials in which the subject did not provide a recognition response, and 2) the question/button-press period. Each regressor was convolved with a gamma-shaped hemodynamic response function (half-width of 3 s and lag of 6 s).

For each subject, a fixed effects analysis modeled the average response to each stimulus condition across runs. Subject-level parameter estimates were entered into a mixed effects analysis (FLAME1 method) to produce group-level estimates of each condition. To localize brain activation and deactivation following presentation of liminal object stimuli, we averaged group-level parameter estimates across stimulus categories, resulting in four contrasts: real objects subjectively recognized > baseline, real objects subjectively unrecognized > baseline, scrambled objects subjectively recognized > baseline, and scrambled objects subjectively unrecognized > baseline. We next computed two comparative contrasts: recognized real objects > unrecognized real objects, and real objects > scrambled objects. We modeled the response to each category separately to avoid confounding recognition responses with category-selective responses when recognition rates differed across categories (Supplementary Fig. 1b). However, in an additional analysis we confirmed that a GLM with only four regressors (rec/unrec x real/scrambled) resulted in a qualitatively similar recognized > unrecognized contrast map, albeit with weaker effect sizes.

Whole-brain, group-level statistical maps for contrasts of interest were thresholded using FSL's FLAME1 cluster-based approach implementing gaussian random field theory to correct for multiple comparisons across voxels, with a cluster-defining voxel threshold of $p < 0.01$ and cluster size $p < 0.05$. FLAME1 has been shown to produce acceptable false positive rates in comparison to other common thresholding methods[94]. In all group analyses, to account for missing voxels in four subjects due to limited field of view or movement during functional MRI acquisition, we calculated group-level statistics in missing voxels by using only those subjects who contained data in these voxels (≥21 out of 25 subjects in all

cases). These voxel statistics were added to the full-brain statistic image before applying cluster-based thresholding. This process increased the number of analyzed voxels by 7.8%.

**ROI definition from recognition contrast**. The recognized > unrecognized GLM contrast revealed one extensive, contiguous cluster showing a positive contrast effect across the brain, while we observed negative effects in several independent clusters. To identify individual clusters that encompass specific brain regions, we applied a more conservative threshold to both positive and negative effects (voxel $p < 0.001$, cluster size $p < 0.05$). The whole-brain map at this stricter threshold identified the same coarse brain areas as the original, more liberal threshold, but with lower significant voxel counts and better separation between neighboring clusters. A few clusters remained that still covered easily separable brain regions. For these, we manually split each cluster by masking with anatomical ROIs from the Harvard-Oxford cortical and subcortical atlas distributed with the FSL analysis package[90]. The clusters that were manually split were: left anterior insula and left middle frontal gyrus; right anterior insula, right middle frontal gyrus, and right orbitofrontal cortex; basal ganglia and thalamus. All of the final clusters were aligned back to each individual subject's anatomical space for use in ROI-based analyses. For reporting cluster locations, we labeled up to 4 local maxima (based on contrast Z values) that best describe the structures that make up each cluster (Supplementary Table 1).

**Visual object localizer**. Subjects viewed images of objects (same as in the main task but full-contrast) and performed a one-back memory task. This task was constructed using a standard block design of twenty 14-second blocks, with 8 s of fixation between blocks. Each block consisted of images from only one category, such that there were 5 blocks for each category randomly interwoven throughout the run. Each block contained 14 trials, and in each trial an image was presented for 500 ms followed by 500 ms blank screen fixation. Subjects were instructed to press any button whenever they saw the same image appear twice in a row. We used a GLM to estimate object category selectivity in higher-order visual cortex. Boxcar regressors were constructed for each category and model parameter estimates were entered into contrasts of faces>others, animals>others, houses>others, manmade objects>others. An anatomical mask of higher-order visual cortex was created by first combining regions taken from the Harvard-Oxford brain atlas: inferior lateral occipital cortex, occipital fusiform gyrus, posterior parahippocampal gyrus, temporal fusiform cortex, temporal occipital fusiform cortex. Next, voxels overlapping with early visual cortical regions V1-V4 (see *Retinotopic mapping*) were excluded. For each GLM contrast, we defined ROI as voxels passing a contrast Z-value threshold of 2.3 within this mask. If less than 100 voxels survived the threshold, we did not define an ROI for that contrast. If the number of significant voxels within a particular contrast was greater than 500, the Z threshold was progressively increased by 0.1 until the voxel count fell below 500. These voxel count cut-offs were determined based on an early study of ventral temporal category selectivity[95] that found average cluster volumes between 900 and 4900 mm³. On average, our category-selective ROIs contained 330.7 ± 16.3 voxels, corresponding to 2645.6 ± 130.4 mm³.

**Retinotopic mapping**. Early visual cortical regions were identified using retinotopic mapping as described in[22]. Briefly, a bar-shaped aperture moved across the screen to reveal a circular checkerboard with 11.78° diameter, with one sweep across the screen lasting 36 s. The bar moved across the screen in 8 sweeps, with each sweep following a different combination of bar orientation and direction. Subjects fixated at a dot at the center of the screen and were instructed to press a button when the dot color changed between green and red. Color changes occurred semi-randomly with approximately two changes per sweep. Preprocessing was performed using AFNI and included motion correction by aligning each volume to the first volume in the run, linear trend removal using linear least-squares, and spatial smoothing (5 mm FWHM kernel). Population receptive field (pRF[96]) analysis was also performed in AFNI[97] to find the receptive field center and size that best explained the observed BOLD time-series for each voxel. The model outputs were X and Y location, sigma (size) and R² estimate of the best-fitting model. X and Y data were converted into polar angle and eccentricity components. In order to manually delineate different visual regions, these components were projected onto each individual subjects' Freesurfer-derived cortical surface reconstructions. Following the guidelines from[97], in both left and right hemispheres of each subject we defined the following visual field maps: V1, V2d, V2v, V3d, V3v and V4. Within each hemisphere, we merged the dorsal and ventral portions of V2 and V3 into one single V2 or V3 ROI. Two subjects' retinotopic data were too noisy for acceptable ROI definition: therefore, all analyses involving early visual cortex ROIs combined data from 23 subjects only.

The first five subjects reported difficulty fixating for the full extent of the retinotopy scan and we observed noisier than expected retinotopic maps. For the remaining 20 subjects, we thus included four additional shorter (~2 min each) retinotopic mapping scans while viewing rotating checkerboard-patterned wedges (to measure polar angle) and expanding/contracting checkerboard-patterned rings (to measure eccentricity). Subjects performed the same color-change detection task as for the first retinotopy scan. Stimuli were presented in the following order:

clockwise rotating wedge, counter-clockwise rotating wedge, expanding ring, contracting ring. All stimuli subtended 5.89° of visual angle at their maxima. Preferred polar angle and eccentricity values for each voxel were derived using the @RetinoProc command in AFNI, and we defined visual field maps using the same method as above. Finally, for each visual area we extracted the larger ROI derived from the two retinotopic mapping methods. On average across subjects and collapsed across hemispheres, retinotopic ROI voxel counts were 883.8 ± 32.8 (V1), 875 ± 36.7 (V2), 741.5 ± 42.4 (V3), 268.1 ± 42.8 (V4).

**Multivariate pattern analysis (MVPA).** We performed MVPA using The Decoding Toolbox[98] in MATLAB within the following ROIs: significant clusters from the group-level recognized>unrecognized GLM contrast, and individually-defined early (retinotopic) and higher-order (category-selective) visual cortex. The two visual ROIs were matched for the number of voxels within each subject, selected according to the following procedure: all subregions were combined to form two larger regional ROIs (V1–V4 combined into early visual cortex, and four category-selective regions combined into higher-order visual cortex). The region with a greater voxel count had voxels removed until it contained an equal number of voxels to the smaller region. The removed voxels were those that had the lowest parameter estimates in a visual object-responsive GLM derived from the object localizer data. Decoding was additionally performed within each visual subregion separately (V1–V4 and four category-selective ROIs) but not controlled for voxel counts.

We trained three linear support vector machine classifiers (cost parameter = 1) to decode stimulus category from GLM-derived beta estimates corresponding to recognized real object trials, unrecognized real object trials, and scrambled object trials. Scrambled image trials were not separated by subjective recognition because recognized, scrambled trial counts were too low for effective classifier training and N-fold cross-validation: subjects frequently recognized only one scrambled image per run. Beta maps in functional space (2 mm isotropic voxels) were aligned to individual subject anatomical space but kept at the 2 mm isotropic voxel resolution. Decoding accuracy was determined using a leave-one-run-out cross-validation scheme. In each cross-validation fold, the classifier was trained using beta estimates from all but one task run and tested on beta estimates from the left out run. Six binary classifications were performed, consisting of all possible pairwise combinations of the four stimulus categories. For each binary classification, the test sample was assigned to one of two categories. Whichever category was chosen in a majority of these binary classifications was selected as the predicted category. To resolve ties, the predicted category was the one that had the maximum decision value summed over all binary classifications[98]. Accuracy was first calculated for each stimulus category separately, and then averaged across categories to produce a final balanced accuracy output per ROI. Significant decoding accuracy at the group-level was assessed using a label permutation test. For each subject, category labels were randomly shuffled within each run separately to maintain cross-validation structure. The classifier was trained and tested using the actual beta estimates but shuffled category labels. This process was repeated 100 times per subject, to generate 100 sets of permuted balanced accuracies. 1000 group-level averages were created by randomly sampling one permuted balanced accuracy output from each subject. The true group-averaged balanced accuracy was tested against this null distribution with α = 0.05, followed by FDR correction (q threshold = 0.05) across all ROIs to account for multiple comparisons. The same permutation test was performed to compare accuracy differences between recognized and unrecognized image trials: the true accuracy difference (recognized accuracy – unrecognized accuracy) was compared to a null distribution of 1000 permuted accuracy differences.

A searchlight-based MVPA analysis was also performed in a similar manner, except a 3-voxel radius sphere centered on each voxel across the whole brain was used instead of ROIs. Statistics were performed using permutations and threshold-free cluster enhancement (TFCE)[99], which accounts for both cluster-like behavior and multiple comparison correction. Briefly, each permuted group average map was tested against chance (25%). Each voxel's t-statistic was transformed using TFCE, and the maximum TFCE value across the brain was incorporated into a null distribution of maximal statistic. The real group average was also TFCE transformed and tested against the null distribution of maximal statistics. Voxels surviving a threshold of p < 0.05 were declared significant.

**Linear mixed model.** We constructed a linear mixed model to assess the effects of ROI voxel count and brain region location (cortical or subcortical) on accuracy of decoding the object category in recognized trials. Using the *lme4* package in R, we fit a model with four-fixed effects parameters: voxel count, brain location, their interaction and the intercept. We additionally included all four parameters as subject-level random effects to ensure maximal random effects structure justified by the design[100]. The model was fit using restricted maximum likelihood estimation (REML) and the BOBYQA optimization algorithm. To improve the model fit, voxel counts were scaled by 1/1000 and centered to zero mean, and decoding accuracies were angular transformed (arcsine($\sqrt{\text{proportion}}$)). The model summary is reported in Supplementary Table 3.

**Reporting summary.** Further information on research design is available in the Nature Research Reporting Summary linked to this article.

## Data availability
Due to the large file size of raw 7 T fMRI data, the datasets generated and analyzed during the current study are available from the corresponding author by reasonable request after appropriate de-identification is carried out. Stimulus images were selected from public domain labeled photographs or from Psychological Image Collection at Stirling (PICS, http://pics.psych.stir.ac.uk/). Source data are provided with this paper.

## Code availability
We used publicly available open source software toolboxes, and custom scripts written in MATLAB, to analyze our data. Code supporting this study is available at a dedicated Github repository: https://github.com/BiyuHeLab/NatCommun_Levinson2021.

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

## Acknowledgements

This research was supported by a National Science Foundation CAREER Award to B.J.H. (BCS-1753218). We thank Siemens Medical Solutions USA, Inc. and the Center for Magnetic Resonance Research at University of Minnesota for sharing the multi-band fMRI sequence. We thank Karina Melnik and Navin Kariyawasam for helping with fMRI data preprocessing, Yuan-hao Wu for comments on the manuscript draft. Graphic elements for figures were downloaded from https://publicdomainvectors.org/.

## Author contributions

B.J.H.: supervision, project administration, funding acquisition; B.J.H. and E.P.: conceptualization; B.J.H., M.L., E.P.: methodology, writing - original draft, writing - review and editing; M.L.: software, formal analysis; M.L. and E.P.: visualization, experimental data collection, data curation; M.L., E.P., S.H.B.: investigation.

## Competing interests

The authors declare no competing interests.
