## [Peer Review File · Nature Communications]

REVIEWER COMMENTS

Reviewer #1 (Remarks to the Author):

Levinson and colleagues investigate the neuronal mechanisms underlying conscious object recognition. Participants viewed object and scrambled stimuli at contrast threshold and reported both the category of the object (face, house, object, animal) and the quality of recognition (yes, no for meaningful object detail perceived). The authors compared activity for recognized and unrecognized stimuli (revealing widely distributed cortical and subcortical responses) and information content using multivariate analyses (revealing decoding in cortical but not subcortical areas). Finally, scrambled images elicited strong activity in prefrontal areas. The authors interpret their results as a challenge for the global neuronal workspace (GNW) theory of consciousness, given widely distributed responses for both recognized and unrecognized stimuli.

This is a well-written and well-presented manuscript – the data are presented clearly, and the analyses are thorough. I have no real concerns about the results themselves, although I think the authors do need to report SNR to help substantiate some of their claims. However, I do have major concerns about the interpretation and the assumptions inherent in the design of the study that I, at least, would like the authors to address.

Major Comments:

1) As I understand the design, the authors are treating the unrecognized trials as reflecting unconscious processing. But I don't think it is trivial to equate a participant's report of unrecognized with unconscious. In this particular design, participants were told to report recognized if they saw meaningful object information. This does not mean that the participants had no conscious percept on unrecognized trials, just that it wasn't meaningful. It's not clear what participants percept was on those trials. Further, it all depends on each participants criterion for what constitutes 'meaningful'. In this context the fact that activity on unrecognized trials was similar, albeit weaker, to recognized trials is perhaps not surprising. Maybe a conscious percept of any stimulus (meaningful or not) would produce such a result? Participants were also expecting a stimulus to be presented, so it's likely that there may be some anticipatory activity, regardless of what happens on any given trial. Since this comparison of recognized and unrecognized trials is being used to argue against aspects of the GNW theory, it is critical they clarify. I suspect the authors do have an answer for this concern (and I recognize it's a concern that applies to much of the literature of consciousness that relies on report), but it's important they lay it out (and I hope I didn't miss it in the manuscript).

2) The authors focus on the failure of category decoding in subcortical regions and I'm concerned they may be over-interpreting the null. One potential interpretation of this result is the presence of weaker signals from these regions compared to cortical areas. The authors claim "equal sensitivity to cortical and subcortical regions" (lines 530-531), but I could find no analysis of signal to noise across the fMRI volumes. On what basis are they claiming equal sensitivity? Although the data were collected at 7T, the voxel size (2 x 2 x 2 mm) is actually quite large for 7T. Physiological noise effects can be large at 7T and most studies collect smaller voxels to mitigate these effects.

Minor Comments:

1) In Figure 3B, the authors acknowledge the circularity of plotting responses from ROIs showing recognized > unrecognized and appropriately state that "statistics were not performed". Yet, they plot the SEM, which is a statistic.

2) Figure 4B – I'm concerned there could be some sort of selection bias affecting the data for recognized and unrecognized trials. Were the ROIs defined using the whole data or just using the training data? If the former, this could be adversely affecting the ability to decode on unrecognized trials.

Reviewer #2 (Remarks to the Author):

In this paper the authors set to examine the wider impact of conscious object recognition on cortical and subcortical regions and large-scale networks. To do so, they have employed liminal object displays with manipulated contrast, 7T fMRI imaging, and decoding of information content. This is a technically sophisticated approach, addressing a still-standing big question. The study was very well conducted, with no comments on the way things were carried out. That said, the final result adds little to the existing literature, unfortunately.

First, there is no bottom-line to speak of here; the results and even the discussion read more like a list of findings that, while may be seen as somewhat interesting individually, do not add up to a clear and important message. Sure, conscious object recognition recruits many cortical and subcortical regions, but why and how remains unclear. Objects entail much more than their identity (e.g., memories and associations, emotions, thoughts about action and affordness). this widespread activation is not really news, and without discerning the widespread activation there is little to be gained.

Second, a worthy goal was to look at the possible involvement of the default-mode network in conscious recognition. Yet, unfortunately, the discussion in this context is superficial and speculative.

It is difficult to write a negative review, especially when so much effort and talent has been invested in this study. Perhaps there is a way for the authors to think harder about a novel and significant story that might be hidden in their rich data.

Reviewer #3 (Remarks to the Author):

Review of "Large-scale cortical and subcortical signatures of conscious object recognition" by Levinson et al.

This is an important paper that brings a wealth of new fMRI data on the neural correlates of conscious perception. It is globally well designed, well analyzed, and goes into every detail of the results. I have no doubt that it will become an important landmark in the stream of recent research on the neural correlates of consciousness in humans.

I do, however, have several important remarks that I think the authors should address before the paper can be accepted. Their paper will only become stronger!

1. In figure 1E it is explicitly mentioned that the very same image can be perceived or unperceived. If so, this would be a wonderful feature of the design. However, I saw no mention of this point later in the manuscript. This is a crucial point, because, for the same value of contrast, it could simply be that some images are easier to recognize than others, and the findings have nothing to do with consciousness per se, but only with "recognizability". Furthermore, a difference between the images could also explain why decodability is much lower for unrecognized images. Again, this would not be a property of conscious perception, but only of the specific images used. This possibility seems all the more plausible that the authors used a measure of contrast which is based on the difference in intensity of the single most white and most black pixels in the image, a value which could be highly affected by noise in the picture. Hence, the noisiest pictures would have been squashed more by the contrast normalization procedure, and would have become even harder to recognize.

To address this point, I suggest

- Showing a histogram of the % recognition across the different pictures, to prove that many pictures are close to 50% (within subjects).
- Redoing the fMRI analysis with regressors restricted to the (subject-specific) pictures that were sometimes recognized and sometimes not recognized.

2. An important result is that the contents of consciousness are not decodable from thalamus and other subcortical areas. However, two important caveats come to mind.

a. This is a null result. Perhaps Bayesian statistics should be used? Or, at the very least, can the authors use a statistic to prove that decodability is significantly better in cortical than in subcortical regions?

b. Could this be due to the smaller number of voxels in subcortical regions? A control analysis could use an identical number of (randomly chosen) voxels in a cortical area – or a cortical area of similar size.

3. Overall, the results provide strong support for the GNW theory of consciousness. In particular, the experiment provides some of the strongest evidence I know of for a critical concept of GNW, broadcasting: the contents of consciousness are decodable from virtually all areas of cortex, including deactivated resting-state regions. This point could be emphasized more, and figure 4B could be improved to show an anatomical diagram of the actual distribution of all ROIs where decodability was significant for recognized trials. Perhaps you could use the same format to show the ROIs where activation is significantly stronger than on unrecognized trials. I would suggest using a complete set of ROIs covering the entire cortical surface (e.g. Van Essen atlas). It would be very interesting to see if virtually all areas, including e.g. tactile or auditory areas also receive the broadcast message, as GNW would predict.

4. The authors are critical of GNW at several places, e.g. “The overlapping spatial extent of brain response between recognized and unrecognized trials is surprising and contrary to the GNW theory, which predicts that conscious perception involves recruitment of additional large-scale cortical networks that are largely silent during unconscious perception” or “Contrary to leading theories of conscious perception, we observed widespread brain responses to both recognized and unrecognized object images with nearly identical spatial extent but different response magnitudes.”

The authors may want to acknowledge that there is, in fact, a straightforward explanation in terms of GNW, ie the lack of temporal resolution of fMRI. With neurophysiological recordings (e.g. Van Vugt et al, Science, 2018), it can be clearly seen that the wave of activation arising from an unseen stimulus can reach prefrontal areas (and other key sites of the GNW) without triggering a full-blow ignition, but only a transient firing that quickly dies out. Such “failed ignition” was simulated and is an integral part of GNW theory since at least the Dehaene & Naccache 2001 paper. Thus, I think that the authors should mention more explicitly both the lack of temporal resolution of the present study, and the tentative explanation of their data in terms of failed ignition. It remains extremely interesting that this early wave may not contain as much decodable information on unrecognized trials than on recognized trials (assuming that the authors perform the above analysis with identical images). This might suggest that the reverberation and amplification that occurs on recognized trials may be used to infer and extract a more detailed picture of the stimulus, as the authors suggest.

5. In fact, since the temporal resolution of fMRI is not nil, I wonder whether the authors could show us BOLD activation (and deactivation!) curves as a function of time for recognized versus unrecognized trials. This analysis might reveal that the activation is more prolonged on recognized trials, as predicted by GNW theory. That would be beautiful!

Again, I would like to reiterate that this is a beautiful study, excellently conducted. The above remarks are only meant as a way to further improve it, and I have no doubt that it should be published.

Minor points

- The authors use the terminology « a binary pattern » to mean that a phenomenon is either present or absent (e.g. decodability). I found this terminology confusing, as “binary” generally refers to the code (at least in computer science). I suggest using the more widespread term « all or none »

We thank the reviewers for their thoughtful and constructive suggestions. We have thoroughly revised the manuscript to fully address all of the questions and concerns raised by the reviewers. Below we respond to each of the reviewers' comments in detail. Blue fonts are quoted text from the reviewers' original comments; black fonts are our replies. Black calibri font with indentation denotes quoted text from the revised manuscript.

Reviewer 1

Levinson and colleagues investigate the neuronal mechanisms underlying conscious object recognition. Participants viewed object and scrambled stimuli at contrast threshold and reported both the category of the object (face, house, object, animal) and the quality of recognition (yes, no for meaningful object detail perceived). The authors compared activity for recognized and unrecognized stimuli (revealing widely distributed cortical and subcortical responses) and information content using multivariate analyses (revealing decoding in cortical but not subcortical areas). Finally, scrambled images elicited strong activity in prefrontal areas. The authors interpret their results as a challenge for the global neuronal workspace (GNW) theory of consciousness, given widely distributed responses for both recognized and unrecognized stimuli.

This is a well-written and well-presented manuscript – the data are presented clearly, and the analyses are thorough. I have no real concerns about the results themselves, although I think the authors do need to report SNR to help substantiate some of their claims. However, I do have major concerns about the interpretation and the assumptions inherent in the design of the study that I, at least, would like the authors to address.

We thank the reviewer for appreciating our work and for the helpful suggestions.

Major Comments:

1) As I understand the design, the authors are treating the unrecognized trials as reflecting unconscious processing. But I don't think it is trivial to equate a participant's report of unrecognized with unconscious. In this particular design, participants were told to report recognized if they saw meaningful object information. This does not mean that the participants had no conscious percept on unrecognized trials, just that it wasn't meaningful. It's not clear what participants percept was on those trials.

The reviewer is correct in that the reports of lack of recognition do not necessarily imply fully unconscious vision. Our intention was to focus on high-level conscious perception (i.e., conscious object recognition) rather than on conscious detection of low-level image features. We have now further clarified this in the text (line 96-100):

“Subjects were instructed to respond “yes” whenever they saw an object, even if the visibility was unclear, and respond “no” when they saw nothing or low-level features only, such as lines or cloud-like abstract patterns. This definition of subjective recognition, allowing unrecognized trials to include conscious perception of low-level features, is consistent with prior studies of conscious object recognition [34, 35, 36].”

34. Grill-Spector K, Kushnir T, Hendler T, Malach R. The dynamics of object-selective activation correlate with recognition performance in humans. *Nature Neuroscience* 3, 837-843 (2000).
35. Fisch L, *et al.* Neural "ignition": enhanced activation linked to perceptual awareness in human

36. ventral stream visual cortex. *Neuron* **64**, 562-574 (2009).
Bar M, *et al.* Cortical mechanisms specific to explicit visual object recognition. *Neuron* **29**, 529-535 (2001).

Further, it all depends on each participants criterion for what constitutes 'meaningful'. In this context the fact that activity on unrecognized trials was similar, albeit weaker, to recognized trials is perhaps not surprising. Maybe a conscious percept of any stimulus (meaningful or not) would produce such a result? Participants were also expecting a stimulus to be presented, so it's likely that there may be some anticipatory activity, regardless of what happens on any given trial. Since this comparison of recognized and unrecognized trials is being used to argue against aspects of the GNW theory, it is critical they clarify. I suspect the authors do have an answer for this concern (and I recognize it's a concern that applies to much of the literature of consciousness that relies on report), but it's important they lay it out (and I hope I didn't miss it in the manuscript).

We thank the reviewer for raising this important issue of whether the spatially overlapping responses to unrecognized and recognized images may partially stem from conscious detection of low-level stimulus features. We now address this potential concern in the revised manuscript (lines 424-432):

"One might wonder whether our result of widespread overlapping responses in recognized and unrecognized trials partially stems from the detection of low-level stimulus features, which could potentially be consciously perceived in unrecognized trials. If this were the case, one would expect to observe above-chance category decoding accuracy in unrecognized trials because low-level features of object stimuli differ across object categories [38, 43]. Our data and experimental procedure are sensitive to such category-specific low-level features of object images, as is evident from successful category decoding from brain responses to scrambled images (Figure 7C). Because the responses to unrecognized real images did not contain any information about object category (Figure 4B), it is unlikely that the widespread activation stems from conscious perception of low-level image features."

38. Podvalny E, Flounders MW, King LE, Holroyd T, He BJ. A dual role of prestimulus spontaneous neural activity in visual object recognition. *Nature communications* **10**, 3910 (2019).
43. Coggan DD, Liu W, Baker DH, Andrews TJ. Category-selective patterns of neural response in the ventral visual pathway in the absence of categorical information. *NeuroImage* **135**, 107-114 (2016).

The reviewer is correct that the classification into 'recognized' and 'unrecognized' depends on the individual subject's criterion. We have now addressed this concern in the revised manuscript under "Limitations and future directions" (lines 566-569):

"Third, following prior studies of conscious object recognition, the "recognized" versus "unrecognized" responses are influenced by individual participants' criteria for a meaningful object percept. While such cross-subject variability is not the focus of the present study, it is of interest for future work to understand individual differences in object perception."

Lastly, we note that our task paradigm and data analyses largely controls for anticipatory brain activity: 1) The stimulus was always presented at central fixation, so spatial anticipation is constant from trial to trial; 2) the prestimulus blank interval was randomly drawn from an exponential distribution between 6 and 20 sec, rendering the stimulus timing unpredictable (i.e., roughly flat hazard rate); 3) the stimulus category was randomly chosen on each trial, rendering stimulus category unpredictable. Therefore, the influence of anticipatory brain activity due to the

subject's active anticipation on our data analyses, which depend on within-subject across-trial comparison, is likely minimal.

2) The authors focus on the failure of category decoding in subcortical regions and I'm concerned they may be over-interpreting the null. One potential interpretation of this result is the presence of weaker signals from these regions compared to cortical areas. The authors claim "equal sensitivity to cortical and subcortical regions" (lines 530-531), but I could find no analysis of signal to noise across the fMRI volumes. On what basis are they claiming equal sensitivity?

We thank the reviewer for raising the question of potential signal sensitivity differences between cortical and subcortical regions. We have now revised the text quoted by the reviewer ("equal sensitivity to cortical and subcortical regions") to "high sensitivity to cortical and subcortical regions" (line 561). We now mention in the revised manuscript the possibility of lower subcortical sensitivity, and have revised the text to provide qualifications to our subcortical chance-level decoding results (lines 476-484):

"It is possible that this null result is partially due to lower fMRI sensitivity to subcortical activity. Currently, no non-invasive neuroimaging method in humans provides equal sensitivity to cortical and subcortical regions. However, prior work demonstrates that 7T fMRI can measure subcortical signals involved in perceptual processes with a considerable improvement over 3T fMRI (De Martino et al., 2018). It remains possible that subcortical regions encode content-specific information in finer-grained activity patterns within a single voxel, which we cannot detect. However, our finding suggests that content-specific information in these subcortical regions is at least much weaker than that in the cortical networks, resulting in undetectable content information in subcortical regions using the current methods involving high-field fMRI, in contrast to the strong content-specific information in widespread cortical networks."

Although the data were collected at 7T, the voxel size (2 x 2 x 2 mm) is actually quite large for 7T. Physiological noise effects can be large at 7T and most studies collect smaller voxels to mitigate these effects.

We would like to point out that it is not the case that decoding accuracy necessarily increases with spatial resolution. In a previous 7T fMRI study using naturalistic movie stimuli (Mandelkow et al., 2017), the authors found that, with data collected at 1.2 mm voxel size, the best decoding accuracy was achieved at a smoothed resolution close to 3 mm, well above the voxel size of data acquisition. The authors explained the reduction of decoding accuracy at high spatial resolution as follows: "*reproducible neuronal contributions were spatially auto-correlated and smooth, while other components of higher spatial frequency were likely related to physiological noise and responsible for the reduced CA [classification accuracy] at higher resolution.*" Based on this finding then, decoding results obtained at a higher spatial resolution could potentially be more susceptible to physiological noise.

Nevertheless, we fully appreciate the reviewer's concern about the possibility of the chosen voxel size affecting decoding performance, and have now addressed this possibility in the revised manuscript (lines 479-480):

"It remains possible that subcortical regions encode object content-specific information in finer-grained activity patterns within a single voxel, which we cannot detect."

Minor Comments:

1) In Figure 3B, the authors acknowledge the circularity of plotting responses from ROIs showing recognized > unrecognized and appropriately state that “statistics were not performed”. Yet, they plot the SEM, which is a statistic.

We thank the reviewer for the suggestion to clarify. The SEM error bars in Fig. 3B are descriptive, to provide readers with insight into the cross-subject variability of the results shown in Fig. 3A. We do not make any statistical inferences from these error bars. We clarified this in the figure legend by changing “statistics were not performed” to “statistical testing was not performed.”

2) Figure 4B – I’m concerned there could be some sort of selection bias affecting the data for recognized and unrecognized trials. Were the ROIs defined using the whole data or just using the training data? If the former, this could be adversely affecting the ability to decode on unrecognized trials.

The reviewer is correct: in the analysis presented in Fig 4B, the ROIs were selected by the recognized vs. unrecognized activation contrast applied to the full data set (see Fig. 3A). This could have introduced a bias in the comparison of decoding accuracy between recognized and unrecognized trials. This concern was addressed by our whole-brain searchlight analysis which does not involve the use of pre-defined ROIs. In this analysis, recognized and unrecognized trials were separately analyzed, and a cross-validation scheme within each group of trials was used to decode stimulus category. Importantly, recognized and unrecognized trials had roughly equal number of trials, because recognition rate was ~50%. We have now further clarified this analysis in the revised manuscript (line 242-248):

“One potential concern with this analysis is that decoding performance could be affected by a bias in ROI selection, given that the ROIs were defined based on differential activation magnitudes between recognized and unrecognized trials. To address this potential issue, we performed a searchlight-based decoding analysis across the whole brain, separately for recognized and unrecognized trials. This analysis confirmed the widespread presence of object category information in recognized trials (Fig. S2A; $p < 0.05$, cluster-based permutation test across the whole brain) and failed to detect any significant category information in unrecognized trials (Fig. S2B), in agreement with the ROI-based results.”

Reviewer 2

In this paper the authors set to examine the wider impact of conscious object recognition on cortical and subcortical regions and large-scale networks. To do so, they have employed liminal object displays with manipulated contrast, 7T fMRI imaging, and decoding of information content.

This is a technically sophisticated approach, addressing a still-standing big question. The study was very well conducted, with no comments on the way things were carried out. That said, the final result adds little to the existing literature, unfortunately.

First, there is no bottom-line to speak of here; the results and even the discussion read more like a list of findings that, while may be seen as somewhat interesting individually, do not add up to a clear and important message. Sure, conscious object recognition recruits many cortical and

sub-cortical regions, but why and how remains unclear.

The reviewer is correct in that figuring out *why* conscious perception at large correlates with certain characteristics of neural activity is an important question. In fact, an answer to this question is one of the ultimate goals of our research on conscious perception. However, one cannot definitively answer the questions of *why and how* without answering the question of *what properties of neural activity support conscious perception*. In the current state of research on conscious perception, we and others are engaged in the iterative process of conceiving possible theories, conducting experiments to test predictions of these theories, and refining the theories when applicable.

We summarize the current state of the field in the Introduction section of the manuscript and explain in the Discussion section how our study adds several important dimensions to the current understanding of conscious object recognition. Here we briefly summarize the main points for the reviewer's convenience:

First, despite intense research, there is currently no general consensus on the brain regions and networks involved in representing the content of conscious experience. Our discovery of significant content decoding across activated and deactivated cortical networks is crucial for the ongoing debate between leading theories of conscious perception. Specifically, our results diverge from theories suggesting that the content of conscious perception is only/primarily encoded in sensory cortices, and we provide some of the first/strongest evidence in support of theories that propose large-scale distributed content representation of conscious perception (e.g., see Reviewer #3's point 3).

Second, we show that the involvement of these large-scale networks is not specific to conscious recognition: the same set of distributed regions are activated or deactivated in response to both recognized and unrecognized images (Fig. 2). Such whole-brain anatomical overlap between perceptual outcomes, with differing response magnitudes (Fig. 3), has not been observed before. This finding provides new constraints on theories of conscious perception, and suggests that instead of focusing on the spatial extent of responses—as past debates have been centered on—the type of information encoded in neural responses may be a more important defining characteristic that distinguishes conscious from unconscious processing. We have now elaborated on this point in the Discussion (line 420-423):

“This result calls into question the previous focus on the spatial extent of brain responses as a defining correlate of conscious perception, and raises instead the importance of content-specificity of such activity. That is, responses to recognized images are both stronger and content-specific (and critically, in both activated and deactivated networks, a point we elaborate on below).”

Third, besides showing activity changes in large-scale networks, we found that almost all recruited cortical regions carry content-specific information during conscious recognition (Fig. 4). By contrast, we were not able to detect significant content-specific information in several subcortical regions that also exhibited differential activation magnitudes to recognized vs. unrecognized trials. This is an important step towards understanding “how” these regions are involved in conscious perception. Deeper mechanistic explanations of these findings—e.g., the underlying circuit-level neuronal computations—are beyond the scope of the present fMRI study. However, our findings open up several previously unknown avenues for further research into the neural implementation underlying conscious object recognition.

Objects entail much more than their identity (e.g., memories and associations, emotions, thoughts about action and affordness). this widespread activation is not really news, and without discerning the widespread activation there is little to be gained.

We agree with the reviewer that object perception involves many factors, including those listed here. Our major contribution is the comparison of neural activity elicited by consciously recognized and unrecognized identical object stimuli, and shedding light on how this neural activity represents high-level content of the stimulus. The entire widespread, object-evoked cortical response is changed during conscious recognition (stronger, Fig. 3, and content-specific, Fig. 4). This suggests that conscious recognition may involve modulation of all processes involved in object perception — an important finding that was not previously known. Discerning how the many facets of object recognition (memory, affordance, etc.) specifically relate to conscious object perception within these large-scale networks is a valuable objective for future work.

Second, a worthy goal was to look at the possible involvement of the default-mode network in conscious recognition. Yet, unfortunately, the discussion in this context is superficial and speculative.

As the reviewer notes, whether the DMN is involved in conscious recognition is an important and so-far unaddressed question. Here, we not only found that the DMN deactivates more strongly to recognized than unrecognized images (when the physical visual input is identical), but observed a striking finding: DMN regions carry strong content-specific information via deactivation patterns when objects are consciously recognized (Figs. 3 & 4). This suggests that instead of encoding task-irrelevant thoughts (and therefore becomes suppressed during more engaging task conditions) as previously thought, the DMN actively encodes perceived stimulus content in its deactivation patterns. We discuss this finding and its implications for theories on conscious perception in the Discussion section dedicated to DMN (lines 439-450).

Our results further reveal an important dissociation between activity magnitudes and information content: stronger information encoding in the DMN can be accompanied by overall lower (i.e., more deactivated) activity level. We discuss the implications of this finding, and its relation to previous similar/related findings in the context of memory encoding and prior knowledge-guided visual perception, in the same Discussion section (lines 451-462).

Lastly, we observed an unexpected finding: DMN regions deactivated more strongly to scrambled than real object images, and also contained significant content-specific information about the category from which the scrambled images were generated. The implication of this result was discussed in the perceptual inference section of the Discussion (lines 535-541):

“Similar results were found in DMN regions: scrambled images elicited stronger deactivation than real object images. As with IFJ and dACC, these DMN regions contain content-specific information about the category from which the scrambled images were generated (Fig. 7C), which strongly correlates with the perceived category during false perception (i.e., false-alarm trials; see Fig. 1G, ‘Scram/Y’). Thus, the amplified deactivation to scrambled images in DMN does not reflect suppression of task-irrelevant activity; instead, the most parsimonious explanation is that the DMN regions are also involved in top-down inferential processing and represent the content of perceptual outcome in their deactivation. This is an exciting hypothesis for future

studies to address.”

Together, we hope that the reviewer can see that we have put forth thoughtful and detailed discussions about our findings regarding the DMN and their relations to past findings as well as implications for future theories.

It is difficult to write a negative review, especially when so much effort and talent has been invested in this study. Perhaps there is a way for the authors to think harder about a novel and significant story that might be hidden in their rich data.

We hope that our responses above help to clarify the novelty and significance of our study.

Reviewer 3

This is an important paper that brings a wealth of new fMRI data on the neural correlates of conscious perception. It is globally well designed, well analyzed, and goes into every detail of the results. I have no doubt that it will become an important landmark in the stream of recent research on the neural correlates of consciousness in humans.

I do, however, have several important remarks that I think the authors should address before the paper can be accepted. Their paper will only become stronger!

We thank the reviewer for appreciating our work and for the thoughtful suggestions.

1. In figure 1E it is explicitly mentioned that the very same image can be perceived or unperceived. If so, this would be a wonderful feature of the design. However, I saw no mention of this point later in the manuscript. This is a crucial point, because, for the same value of contrast, it could simply be that some images are easier to recognize than others, and the findings have nothing to do with consciousness per se, but only with “recognizability”. Furthermore, a difference between the images could also explain why decodability is much lower for unrecognized images. Again, this would not be a property of conscious perception, but only of the specific images used.

This possibility seems all the more plausible that the authors used a measure of contrast which is based on the difference in intensity of the single most white and most black pixels in the image, a value which could be highly affected by noise in the picture. Hence, the noisiest pictures would have been squashed more by the contrast normalization procedure, and would have become even harder to recognize.

The images were not all assigned the same contrast: the contrast for each image was determined based on each subject’s threshold for conscious recognition of that particular image. We described this in the original manuscript (revised lines 100-102):

“To identify the liminal image contrast, we conducted an adaptive staircase procedure whereby the contrast of each unique image was titrated to reach a ~50% subjective recognition rate across identical trials for each participant.”

To address this point, I suggest

- Showing a histogram of the % recognition across the different pictures, to prove that many pictures are close to 50% (within subjects).
- Redoing the fMRI analysis with regressors restricted to the (subject-specific) pictures that were sometimes recognized and sometimes not recognized.

Because the contrast staircase procedure was performed separately for each image, we do not expect variability across images to induce any differences in recognition rates. However, as with any study of perceptual behavior involving stimulus uncertainty and a finite set of experimental trials, there can be measured differences in recognition rates across images. We have now implemented the reviewer’s suggestion to include a histogram (Fig. S1A) showing that the vast majority of images were recognized at an average rate of $46.8 \pm 4.7\%$, and that individual stimuli were only very rarely 100% recognized or 0% recognized. We address the issue in lines 116-119 of the revised manuscript:

“We computed every subject’s recognition rates for each individual object image (Fig. S1A). Subjects recognized the majority of object images at a rate of $46.8 \pm 4.7\%$ (mean \pm s.e.m. of the mode, where each subject’s mode is the center of their tallest histogram bin shown in Fig. S1A), confirming success of our staircase procedure on the single-subject level.”

Fig. S1 (reproduced from manuscript). **A:** Histograms of image-specific recognition rates (% of trials reported as recognized) for each subject. Recognition rates were combined into 5 bins of width 20. On average, subjects recognized most images in $46.8 \pm 4.7\%$ of trials (mean \pm s.e.m. of the mode, where each subject’s mode is the center of their tallest bin), indicating threshold-level perceptual variability. n_0 = number of images never recognized

(0%); n_{100} = number of images always recognized (100%).

Given that the vast majority of images were “sometimes recognized and sometimes not recognized” by individual subjects (see Fig. S1A insets for the number of images that are exceptions to this at the individual-subject level), we think that re-doing the fMRI analysis as suggested by the reviewer would not result in any meaningful change.

2. An important result is that the contents of consciousness are not decodable from thalamus and other subcortical areas. However, two important caveats come to mind.

a. This is a null result. Perhaps Bayesian statistics should be used? Or, at the very least, can the authors use a statistic to prove that decodability is significantly better in cortical than in subcortical regions?

b. Could this be due to the smaller number of voxels in subcortical regions? A control analysis could use an identical number of (randomly chosen) voxels in a cortical area – or a cortical area of similar size.

The subcortical regions did not have lower voxel counts than the cortical regions. We described this in the original text (revised lines 224-226):

“The failure of decoding in subcortical regions was not due to differential ROI sizes, as their sizes lie within the range of cortical ROIs (see Table S1 for voxel count for all ROIs).”

Please see the scatter plot below for an additional visualization of the relationship between voxel count and decoding accuracy in recognized trials (taken from Figure 4B): each subcortical area was larger than several cortical areas that showed significant decoding, and the ROI sizes of subcortical regions completely lie within the distribution of ROI sizes of cortical regions that had significant decoding.

In response to Reviewer #1’s major point 2, We have revised the manuscript to include alternative explanations of the null subcortical decoding result, related to limitations of the fMRI methodology (lines 476-484):

“It is possible that this null result is partially due to lower fMRI sensitivity to subcortical activity. Currently, no non-invasive neuroimaging method in humans provides equal sensitivity to cortical and subcortical regions. However, prior work demonstrates that 7T fMRI can measure subcortical signals involved in perceptual processes with a considerable improvement over 3T fMRI (De Martino et al., 2018). It remains possible that subcortical regions encode content-specific information in finer-grained activity patterns within a single voxel, which we cannot detect. However, our finding suggests that content-specific information in these subcortical regions is at least much weaker than that in the cortical networks, resulting in undetectable content information in subcortical regions using the current methods involving high-field fMRI, in contrast to the highly significant content-specific information in widespread cortical networks.”

We feel that it would not be appropriate to directly test cortical vs. subcortical decoding given the wide range of decoding accuracies across cortical regions, and the fact that subcortical and cortical regions may have qualitatively different information processing structures that may require invasive methodologies to further investigate.

3. Overall, the results provide strong support for the GNW theory of consciousness. In particular, the experiment provides some of the strongest evidence I know of for a critical concept of GNW, broadcasting: the contents of consciousness are decodable from virtually all areas of cortex, including deactivated resting-state regions. This point could be emphasized more, and figure 4B could be improved to show an anatomical diagram of the actual distribution of all ROIs where decodability was significant for recognized trials.

We thank the reviewer for the suggestion to more clearly state how our results are compatible with the GNW’s concept of broadcasting. We have revised the relevant section of the Discussion to emphasize this point (lines 406-412; new text in bold):

“Our results during subjective recognition are broadly consistent with the global neuronal workspace (GNW) theory, which predicts that **conscious contents are broadcast throughout a distributed brain network**. Evidence for perceptual content representation in this distributed network during conscious but not unconscious processing was previously weak; **our findings substantially strengthen the evidence for global broadcasting as a correlate of conscious perception**. Specifically, we show that neural representations of content exist during subjective recognition in PFC, ACC, PCC — brain areas proposed by the GNW theory as hubs for information broadcasting during conscious processing [5].”

5. Mashour GA, Roelfsema P, Changeux JP, Dehaene S. Conscious Processing and the Global Neuronal Workspace Hypothesis. *Neuron* **105**, 776-798 (2020).

We address the visualization of Figure 4B below.

Perhaps you could use the same format to show the ROIs where activation is significantly stronger than on unrecognized trials. I would suggest using a complete set of ROIs covering the entire cortical surface (e.g. Van Essen atlas). It would be very interesting to see if virtually all areas, including e.g. tactile or auditory areas also receive the broadcast message, as GNW would predict.

The anatomical distribution of ROIs in Fig. 4B is already visualized in Fig. 3A, which was

explained in the text (lines 216-219). We have now also clarified this in Fig. 4B legend. We thus do not think that it is necessary to reproduce Fig. 3A within Fig. 4.

We agree that it is important to see where content may be broadcast across the whole brain. We believe that our searchlight analysis presented in Figure S2, where we perform the decoding analysis across the entire cortical surface, should address this question. These results are explained in the manuscript (lines 244-251):

“we performed a searchlight-based decoding analysis across the whole brain, separately for recognized and unrecognized trials. This analysis confirmed the widespread presence of object category information in recognized trials (Fig. S2A) and failed to detect any significant category information in unrecognized trials (Fig. S2B), in agreement with the ROI-based results. Notably, the searchlight analysis did not reveal significant stimulus category information in brain areas that did not show BOLD response amplification, suggesting that the distributed category information is contained within those areas showing amplified activation or deactivation during successful recognition.”

As can be inferred from the quoted text above, and seen directly in Fig. S2A, we did not find significant content decoding in recognized trials in tactile or auditory areas. However, we could not find mentioning of this specific prediction in GNW theory papers; therefore we have opted not to comment on this null finding in the manuscript.

4. The authors are critical of GNW at several places, e.g. “The overlapping spatial extent of brain response between recognized and unrecognized trials is surprising and contrary to the GNW theory, which predicts that conscious perception involves recruitment of additional large-scale cortical networks that are largely silent during unconscious perception” or “Contrary to leading theories of conscious perception, we observed widespread brain responses to both recognized and unrecognized object images with nearly identical spatial extent but different response magnitudes.”

The authors may want to acknowledge that there is, in fact, a straightforward explanation in terms of GNW, ie the lack of temporal resolution of fMRI. With neurophysiological recordings (e.g. Van Vugt et al, Science, 2018), it can be clearly seen that the wave of activation arising from an unseen stimulus can reach prefrontal areas (and other key sites of the GNW) without triggering a full-blow ignition, but only a transient firing that quickly dies out. Such “failed ignition” was simulated and is an integral part of GNW theory since at least the Dehaene & Naccache 2001 paper. Thus, I think that the authors should mention more explicitly both the lack of temporal resolution of the present study, and the tentative explanation of their data in terms of failed ignition. It remains extremely interesting that this early wave may not contain as much decodable information on unrecognized trials than on recognized trials (assuming that the authors perform the above analysis with identical images). This might suggest that the reverberation and amplification that occurs on recognized trials may be used to infer and extract a more detailed picture of the stimulus, as the authors suggest.

We thank the reviewer for raising concerns about how our results relate to the GNW theory, which is a critically important topic for the present study. We have rephrased the section describing how our results depart from GNW in lines 413-422:

“The GNW theory suggests that unconscious processing involves a weak, slow-decaying wave of activation along the feedforward pathways, potentially reaching prefrontal cortex, while only

conscious processing triggers global activation ('ignition') involving a more extensive network of reverberant loops, which enables access of a given piece of information to a wider array of brain regions—especially frontoparietal areas^{5, 47, 54}. While our results show non-content-specific activation as a correlate of unconscious processing of objects, it is difficult to explain the wide spatial extent of these responses, nearly identical to that of the responses to recognized images, with a purely feedforward propagation thesis. This result calls into question the previous focus on the spatial extent of brain responses as a defining correlate of conscious perception, and raises instead the importance of content-specificity of such activity.”

We have now addressed the possibility that we missed a “failed ignition’ due to the poor temporal resolution of fMRI in lines 562-565 of the revised manuscript:

“However, due to the slow hemodynamic response, this technique cannot capture fast neural dynamics. We cannot, for example, address whether unrecognized trials contain evidence for the GNW notion of “failed ignition” [14], in which transient and weak content-specific activation fails to ignite the global workspace [5].”

5. Mashour GA, Roelfsema P, Changeux JP, Dehaene S. Conscious Processing and the Global Neuronal Workspace Hypothesis. *Neuron* **105**, 776-798 (2020).
14. Salti M, Monto S, Charles L, King JR, Parkkonen L, Dehaene S. Distinct cortical codes and temporal dynamics for conscious and unconscious percepts. *eLife* **4**, (2015).

5. In fact, since the temporal resolution of fMRI is not nil, I wonder whether the authors could show us BOLD activation (and deactivation!) curves as a function of time for recognized versus unrecognized trials. This analysis might reveal that the activation is more prolonged on recognized trials, as predicted by GNW theory. That would be beautiful!

Due to the trial design (Fig. 1D) and the slow hemodynamic response, any prolonged activation would be masked by the BOLD response to the question/report period beginning 4-6 seconds following stimulus onset. To confirm, we have plotted the BOLD response curves across time in each ROI (see figure i below). While the response magnitudes differ between recognized and unrecognized (as expected given our analysis results), there is no discernible prolonged activation in recognized trials. This is potentially due to the limited temporal resolution of fMRI, as the reviewer suspected.

Figure i. BOLD % signal change from baseline (1 TR before stimulus onset) averaged across trials and subjects. Each subplot refers to one ROI listed in Figure 3B. Data points indicate the midpoint of each volume acquisition. Vertical black lines indicate stimulus onset. Shaded areas indicate standard error of the mean across subjects.

Again, I would like to reiterate that this is a beautiful study, excellently conducted. The above remarks are only meant as a way to further improve it, and I have no doubt that it should be published.

Thank you! We very much appreciate the helpful comments and hope that the reviewer finds that our revisions have further improved the manuscript.

Minor points

- The authors use the terminology « a binary pattern » to mean that a phenomenon is either present or absent (e.g. decodability). I found this terminology confusing, as “binary” generally refers to the code (at least in computer science). I suggest using the more widespread term « all or none »

We thank the reviewer for the suggestion to clarify, and have now changed the text from “binary” to “all-or-none.” (lines 266 and 436)

References

1. De Martino F, Yacoub E, Kemper V, Moerel M, Uludağ K, De Weerd P, Ugurbil K, Goebel R, Formisano E. The impact of ultra-high field MRI on cognitive and computational neuroimaging. *NeuroImage* **168**, 366-382 (2018).
2. Mandelkow H, de Zwart JA, Duyn JH. Effects of spatial fMRI resolution on the classification of naturalistic movies. *Neuroimage* **162**, 45-55 (2017).

REVIEWER COMMENTS

Reviewer #1 (Remarks to the Author):

The authors have fully addressed all the concerns I raised and made appropriate changes to the manuscript. This is a comprehensively presented and analysed study that I'm sure will generate much discussion within the literature.

Reviewer #2 (Remarks to the Author):

I am impressed by the thoughtful revision. It is a sound paper with interesting findings for those in the field.

Reviewer #3 (Remarks to the Author):

After reading the revised paper and the rebuttal letter, I am fully satisfied with the authors' responses. I am sorry that I had missed the information on two critical points – the fact that every image was carefully titrated for 50% recognition; and the control for the number of voxels in the comparison of cortical versus subcortical decoding. These points merely show that the study was conducted even more cleverly than I had thought!

I have only one small quibble. The figure on page 9 of the rebuttal letter is fully convincing in suggesting that decoding is systematically better in cortical than in subcortical areas, regardless of the number of voxels. Why not include this figure as supplementary materials? And why not turn it into statistics? You could do a multiple regression on the decoding scores, with number of voxels and cortical/subcortical as two factors, and show that there is a significant difference due to the latter factor, even when the former (number of voxels) is controlled for.

I am insisting on this point because the current paper is just concluding with something like "significant decoding in cortical, not significant in subcortical" – and as we all know, this is not sufficient to conclude that there is a significant DIFFERENCE between the two (it could be that the subcortical decoding is just below threshold, and the cortical decoding is just above threshold).

This is what I was trying to convey in my previous review: instead of being content with a null effect (decoding in subcortical regions), you could turn it into a (presumably) significant difference between cortical and subcortical regions, thus avoiding to base your conclusions on acceptance of the null hypothesis, which is always a rather weak argument.

Again, congratulations for a beautiful paper.

We thank the reviewers for their kind reception of the revised manuscript. We have further revised the manuscript to address Reviewer 3's final concerns. Blue fonts are quoted text from the reviewers' original comments; black fonts are our replies. Black calibri font with indentation denotes quoted text from the revised manuscript.

Reviewer #1 (Remarks to the Author):

The authors have fully addressed all the concerns I raised and made appropriate changes to the manuscript. This is a comprehensively presented and analysed study that I'm sure will generate much discussion within the literature.

Reviewer #2 (Remarks to the Author):

I am impressed by the thoughtful revision. It is a sound paper with interesting findings for those in the field.

Reviewer #3 (Remarks to the Author):

After reading the revised paper and the rebuttal letter, I am fully satisfied with the authors' responses. I am sorry that I had missed the information on two critical points – the fact that every image was carefully titrated for 50% recognition; and the control for the number of voxels in the comparison of cortical versus subcortical decoding. These points merely show that the study was conducted even more cleverly than I had thought!

I have only one small quibble. The figure on page 9 of the rebuttal letter is fully convincing in suggesting that decoding is systematically better in cortical than in subcortical areas, regardless of the number of voxels. Why not include this figure as supplementary materials? And why not turn it into statistics? You could do a multiple regression on the decoding scores, with number of voxels and cortical/subcortical as two factors, and show that there is a significant difference due to the latter factor, even when the former (number of voxels) is controlled for.

I am insisting on this point because the current paper is just concluding with something like "significant decoding in cortical, not significant in subcortical" – and as we all know, this is not sufficient to conclude that there is a significant DIFFERENCE between the two (it could be that the subcortical decoding is just below threshold, and the cortical decoding is just above threshold). This is what I was trying to convey in my previous review: instead of being content with a null effect (decoding in subcortical regions), you could turn it into a (presumably) significant difference between cortical and subcortical regions, thus avoiding to base your conclusions on acceptance of the null hypothesis, which is always a rather weak argument.

Again, congratulations for a beautiful paper.

Stanislas Dehaene

We have implemented the reviewer's suggestion to conduct a multiple regression on the decoding results, as described in the revised text (lines 224 – 232):

"To test whether there is a difference in decoding accuracy between cortical and subcortical regions and whether it is confounded by voxel count, we fit a linear mixed model (see Methods) with parameters including: ROI location (cortical or subcortical), voxel count, their interaction, in

addition to the intercept and the subject-level random effects for each parameter (model summary is reported in Table S3, group-level data are plotted in Fig. S2E). We observed a significant effect of ROI location (beta = 0.052 ± 0.012, estimate ± s.e.m., p < 0.001), indicating weaker decoding for subcortical regions, but no significant effect of voxel count (beta = 0.011 ± 0.012, p = 0.58) or their interaction (beta = 0.019 ± 0.019, p = 0.31). This result suggests that decoding accuracy is significantly higher in cortical than subcortical ROIs, with voxel count controlled for.”

We describe the modeling procedure in the revised Methods section (lines 845 - 852):

“**Linear Mixed Model.** We constructed a linear mixed model to assess the effects of ROI voxel count and brain region location (cortical or subcortical) on accuracy of decoding the object category in recognized trials. Using the *lme4* package in R, we fit a model with four fixed effects parameters: voxel count, brain location, their interaction and the intercept. We additionally included all four parameters as subject-level random effects to ensure maximal random effects structure justified by the design¹⁰². The model was fit using restricted maximum likelihood estimation (REML) and the BOBYQA optimization algorithm. To improve the model fit, voxel counts were scaled by 1/1000 and centered to zero mean, and decoding accuracies were angular transformed ($\arcsine(\sqrt{proportion})$). The model summary is reported in Table S3.”

The model summary (Table S3) is reproduced below.

Linear mixed model: Recognized category decoding accuracy

Predictors	Estimates	std. Error	CI	p
(Intercept)	0.53	0.01	0.50 – 0.55	<0.001
Cortical [TRUE]	0.05	0.01	0.03 – 0.08	<0.001
Voxel_count	0.01	0.02	-0.03 – 0.05	0.577
Cortical [TRUE] * Voxel_count	0.02	0.02	-0.02 – 0.06	0.314

Random Effects

σ^2	0.01
τ_{00} Subject	0.00
τ_{11} Subject.factor(Cortical)TRUE	0.00
τ_{11} Subject.Voxel_count	0.00
τ_{11} Subject.factor(Cortical)TRUE:Voxel_count	0.00
ρ_{01}	0.07
	0.62
	-0.74
ICC	0.31

N_{subject}	25
Observations	600
Marginal R^2 / Conditional R^2	0.111 / 0.382
log-Likelihood	601.704

Table S3. Effects of ROI location (cortical or subcortical) and voxel count on category decoding accuracy in recognized trials. Recognized object category decoding accuracy was significantly predicted by ROI location (cortical or subcortical), but not ROI voxel count or their interaction, according to a linear mixed model with maximal random effects structure. σ^2 : residual (within-subject) variance, τ_{00} : random intercept (between-subject) variance, τ_{11} : random slope (between subject) variance, ρ_{01} : random slope-intercept correlation, ICC: intraclass correlation coefficient. Marginal R^2 includes only fixed effects variance, while Conditional R^2 includes both fixed and random effects variance.

We additionally included the scatter plot of decoding accuracy by voxel count in Figure S2E. The figure is reproduced below.

Fig. S2E. Relationship between category decoding accuracy and ROI voxel counts, separated into cortical and subcortical regions. Data taken from Figure 4B. Each data point refers to one ROI; error bars denote s.e.m. across subjects. Filled circles indicate decoding accuracies significantly above chance level as assessed by permutation tests. A linear mixed model identified a significant effect of ROI location (cortical or subcortical) on decoding accuracy ($p < 0.001$), but no significant effect of voxel count or significant interaction between ROI location and voxel count (see Table S3).

REVIEWERS' COMMENTS

Reviewer #3 (Remarks to the Author):

All of my remarks have been very carefully considered. I have no more comments, I think that the paper can be accepted as is.